



# Future dust concentration over the Middle East and North Africa region under global warming and stratospheric aerosol intervention scenarios

Seyed Vahid Mousavi[1], Khalil Karami[2], Simone Tilmes[3], Helene Muri[4], Lili Xia[5], Abolfazl Rezaei[1,6]

[1]Center for Research in Climate Change and Global Warming, Institute for Advanced Studies in Basic Sciences, Zanjan, Iran.
[2]Leipzig Institute for Meteorology, University of Leipzig, Germany
[3]National Center for Atmospheric Research, Boulder, CO, USA.
[4]Industrial Ecology Programme, Department of Energy and Process Engineering, Norwegian University of Science and Technology, Trondheim, Norway.
[5]Department of Environmental Sciences, Rutgers University, New Brunswick, USA.
[6]Departmant of Earth Sciences, Institute for Advanced Studies in Basic Sciences, Zanjan, Iran.

*Correspondence to*: Seyed Vahid Mousavi (v.mousavi33@email.com)

**Abstract.** The Middle East and North Africa (MENA) is the dustiest region, in the world and understanding the projected changes in the dust concentrations in the region is crucially important. Stratospheric aerosol injection (SAI) geoengineering aims to reduce global warming, by increasing the reflection of a small amount of the incoming solar radiation to space, and hence reducing the global surface temperatures. Using the output from the Geoengineering Large Ensemble Project (GLENS) project, we show a reduction in the dust concentration in the MENA region under both global warming (RCP8.5) and GLENS-SAI scenarios compared to the present-day climate. This reduction over the MENA region is stronger under the SAI scenario, while for dry season (e.g., summer with the strongest dust events), more reduction has been projected for the global warming scenario. The maximum reduction of the dust concentrations in the MENA region (under both the global warming and SAI) is due to the weakening of the dust hotspots emissions from the sources of the Middle East. Further analysis of the differences in the surface temperature, soil water, precipitation, leaf area index, and near surface wind speed provides some insights into the underlying physical mechanisms that determine the changes in the future dust concentrations in the MENA region. We also conduct wavelet analysis using the time series of the monthly, seasonal, and annual climate changes under the SAI simulation to identify the dust relationship with the considered variables. Our findings show that a stronger reduction of the dust concentration in the MENA region under SAI relative to the RCP8.5 scenario is a complex interplay with temperature reduction, precipitation, soil water and leaf area index enhancement, as well as weakening of near surface winds compared to the present-day climate.



# 1 Introduction

Dust aerosols have a great potential to influence the Earth's climate system (Alpert et al., 1998; Middleton et al., 2017; Wang et al., 2017; Kok et al., 2018), including directly scattering of short-wave radiation, absorption of long-wave radiation (Dufresne et al., 2002; Albani et al., 2014; Mahowald et al., 2014; Kok et al., 2017), and indirectly changing cloud properties and precipitation rates through aerosol-cloud interaction (Atkinson et al., 2013; Sagoo et al., 2017). Furthermore, dust deposition in different environments (particularly on ice and snow) may affect the surface albedo (Krinner et al., 2006; Painter et al., 2013; Albani et al., 2018; Sarangi et al., 2020). Mineral dust may also be transported a long distance and affects areas apart from the emission source, such as the biogeochemistry of the oceans, and hence induces feedbacks within the climate system (Jickells et al., 2005; Cao et al., 2005; Gasso et al., 2010; Wang et al., 2017; Kok et al., 2018). The dust storms can further influence the human health, agriculture, and transport sectors, particularly in the arid and semi-arid regions (Alboghdady et al., 2016; Sternberg et al., 2017). In the Northern Hemisphere (NH), the Middle East and North Africa (MENA) regions, including Sahara and the Middle East, are important sources for the dust emission. The MENA region is part of the NH "dust belt", which extends from North Africa to East Asia which is evident from satellite observations (Ginoux et al., 2012). Generally, the MENA region is dry with a weak and scattered vegetation coverage, partially because it is away from the storm-track regions, and cannot receive humidity transferred from remote (Karami et al., 2019). The MENA region also accounts for the dustiest region in the world (i.e., northern Chad) and the largest warm desert (Sahara) (Giles et al., 2005). Therefore, it is essential to understand dust concentration changes in this region under different future climate scenarios.

Previous researches using different methods and approaches indicate great uncertainty in determining the future changes of the global dustiness. As an example, Tegen et al. (2004) using HADCM3 and ECHAM4 models with IS92a IPCC scenario demonstrated that future dust emissions may increase or decrease. Woodward et al. (2005) using the HadAM3 AGCM model with the IS92a scenario, indicated an increase in the future global dust emissions, while Mahowald et al. (2003) using the National Center of Atmospheric Research's coupled Climate System Model (CSM) 1.0 model with 6 different scenarios suggested a 20–60% reduction in future dust emissions. Under Representative Concentration Pathway (RCP8.5) scenario and using regional climate model (RegCM4.0) (Giorgi et al. 2012), reginal projections over West Africa projected an increase in mineral dust with significant increase over Sahel and Sahara in the warm season (Ji et al., 2015), and Liu et al., (2020) using the fifth Climate Models Intercomparison Project (CMIP5) projected a reduction in the dust events over Northern China under RCP8.5 scenario. The incorporated dust emission, transport, and both dry and wet depositions (collectively called the dust cycle) are incorporated into climate models and Earth system models which greatly differ in dust emission schemes, vegetation cover for dust emission (either prescribed or prognostic) and assumptions about dust sizes (Wu et al., 2020). While most models have the skill to generate the general patterns of global dust distribution (Liu et al, 2012; Huneeus et al, 2011), however there still exist large uncertainties in the simulated global dust budgets estimated by the model results (Huneeus et al., 2011; Textor et al., 2006) which impede the interpretation of





the evolution of dust storms under future climate projections (Boucher et al, 2013; Yue et al., 2010). For example, the recent study of Wu et al (2020) analysed 15 models participated in the CMIP5 project and compared them with an aerosol reanalysis as well as station observations and concluded that while the models generally agree with each other and also with observations in producing the NH dust belt, the models greatly differ in the spatial extent of the dust belt and have large biases in dust deposition regions for some models.

Slow progresses in decoupling anthropogenic emissions from economic growth and negative emission technologies are the main reasons for continuation of the increase in the global atmospheric greenhouse gas concentration (Fuss et al., 2014; Rozenberg et al., 2015; Sanderson et al., 2016). Moreover, previous researches indicated that the current pledges to reduce greenhouse gas emissions would not be sufficient to limit temperature rise beyond 1.5-2°C (compared to the pre-industrial period) (Millar et al., 2017; Pasztor and Turner., 2018). Geoengineering is considered as the third pillar of climate change policy (alongside mitigation and adaptation efforts) to compensate for anthropogenic warming (e.g., Nurse, 2011; Macnaghten and Szerszynski., 2013), and Stratospheric Aerosol Injection (SAI) geoengineering is one of the most discussed strategies. In other words, SAI is an interim measure to offset warmings while the emissions are reduced. Among various geoengineering approaches, SAI has received a particular attention for mainly two reasons a) volcano eruptions may serve as a natural analogy for this strategy, and b) all modelling studies show an efficient global cooling effect with the SAI strategy (e.g., Caldeira and Matthews, 2007; Robock et al., 2008; McClellan et al., 2010; Tilmes et al., 2018; Simpson et al., 2019; Visioni et al., 2020). Climate models may simulate SAI by injecting sulfate aerosols or their precursor (sulfur dioxid, $SO_2$) into the stratosphere, which reflects some of the incoming sunlight back to space (Crutzen, 2006; Rasch, 2008). Other types of aerosols are also being investigated, e.g., the sensitivity of the chemistry-climate models to injection of $H_2SO_4$ instead of $SO_2$ have been investigated (Keith et al., 2019; Vattioni et al., 2020).

There are many unknowns regarding the SAI and its modelling, particularly its potential side effects on regional and local scales. While some of the debates is focused on the technical, financial and even political feasibility of such climate intervention scenarios, the lack of knowledge on the potential impacts, including dust concentration change, of such interventions in local scale is still a main cause for concern (Karami et al., 2020). Therefore, there is an immediate need for knowledge on the dust concentrations response to the possible future climate change scenarios in the MENA region. This might assist to inform the local governments and public of the potential impacts of such climate intervention scenarios. Here, we use the data generated by the Geoengineering Large Ensemble Project (GLENS) project (Tilmes et al., 2018) to (1) examine the future changes of dust concentration in the MENA region under RCP8.5 and SAI, and (2) demonstrate the dust relationship with hydro-climate variables of temperature, soil water, precipitation, leaf area index, and near surface wind. The paper is structured as follows: the method and data are presented in Section 2, the results in Section 3, the discussion in Section 4, and the conclusions are finally drawn in Section 5.



## 2 Data and Methods

In this study, we use the GLENS project output. The GLENS project investigates the impacts of SAI within the climate
variability on the global and regional scale with large ensemble members to reach multiple temperature targets using a
feedback algorithm (Tilmes et al., 2018). GLENS includes a 20-member ensembles of the baseline RCP8.5 scenario for
the period 2010-2030, which serves as a control dataset (hereafter the present-day climate or CTL simulation). Three of
the control simulations were continued until 2097, which serves as the baseline simulation. In addition, there are 20

ensemble members of the SAI simulations from 2020 to 2097.  Annually varying $SO_2$ injections were performed at four
locations (30°N, 30°S, 15°N, and 15°S) into the stratosphere (roughly about 5 km above the tropopause) (Kravitz et al.,
2017), based on a feedback-control algorithm to keep a) the global surface temperature, b) interhemispheric and c) equator-
to-pole temperature gradients close to the year 2020 conditions. In the GLENS project, an updated version of the
Community Earth System Model CESM Version 1 (Hurrell et al., 2013) with the Whole Atmosphere Community Model

(WACCM) as its atmospheric component (Mills et al., 2017) is used. The baseline scenario is the Representative
Concentration Pathway 8.5 (RCP8.5) (2010-2097). The SAI scenario (2020-2097) is based on the same baseline emission
pathway, and uses sulfur dioxide injections to keep surface temperatures at 2020 conditions. The model simulations are
performed with horizontal resolution of 0.9° latitude by 1.25° longitude, and 70 vertical layers up to 140 km (~$10^{-6}$ hPa).
The three-mode version of the Modal Aerosol Module (MAM3) is used to simulate microphysical processes of the aerosols

in the troposphere and stratosphere (Liu et al. 2012) and include prognostic stratospheric aerosols (Mills et al., 2016).
WACCM is fully coupled with the Community Land Model, version 4.5 (CLM4.5) (Oelson et al. 2013) as well as with
other CESM1 components, are listed in the Table 1. Details of simulations, coupled models, and parametrization are
further described by previous researchers (e.g., Danabasoglu et al., 2012; Holland et al., 2012; Guenther et al., 2012;
Marsh et al., 2013; Milles et al., 2016; Milles et al., 2017; Tilmes et al., 2018). In the CLM model, Dust Entrainment and

Deposition model (DEAD) (Zender et al., 2003) is used for atmospheric dust mobilization scheme (Mahowald et al. 2006;
Oleson et al., 2013). Based on the DEAD, the total vertical mass flux of dust (Fj) from the ground into transport bin j is
computed by

$$F_j = TSf_m \alpha Q_s \sum_{i=1}^{I} M_{i.j} \tag{1}$$

where $T$ is a global factor, $S$ the source erodibility factor, $f_m$ grid cell fraction of exposed bare soil suitable for dust

mobilization, $\alpha$ sandblasting mass efficiency, $Q_s$ the total horizontally saltating mass flux of "large" particles, and $M_{i,j}$ the
mass fraction of each source mode $i$ carried in each of $J = 4$ transport bins $j$.

The value of $f_m$ factor and ability of dust to mobilize are highly decreased by increasing the total water content (including
lake, wetlands, and soil moisture) as well as the fraction of vegetation cover in each grid cell (Oelsen et al., 2013), based
on $f_m = (1 - f_{lake} - f_{wetl})(1 - f_{sno})(1 - f_v)w_{liq,1}/(w_{liq,1} + w_{ice,1})$ where $f_{lake}, f_{wetl}, f_{sno}$ and $f_v$ are the grid cell

fractions of lake, wetland, snow cover, and vegetation cover, respectively. $w_{liq,1}$ and $w_{ice,1}$ represent the top soil layer





liquid water and ice contents, respectively. In practice, soil moisture controls the threshold wind friction speed for saltation. Further, the total horizontally saltating mass flux ($Q_s$) is related to the third power of the wind speed; thus, any changes in wind speed could affect the dust emissions (Tegen and Fung, 1994; Tegen et al., 2002; Zender et al., 2003; Oelsen et al., 2013) with a positive correlation. Previous studies also show that a higher vegetation coverage leads to smaller dust

emissions as the vegetation coverage can trap soil moisture through its roots and shade and also reduce soil erosion by reducing wind friction (Hillel et al., 1982; Raupach et al., 1994; Nicholson et al., 1998; Zender et al., 2003). In other words, the total Leaf area index (TLAI) have a negative correlation with dust emissions and subsequently with atmospheric dust which is also depicted in the result section. The dust model also consists of removing mineral dust from the atmosphere, including dry deposition and wet deposition. Wet deposition removes dust aerosols through in-cloud and

below-cloud precipitation process (Albani et al., 2014; Zender et al., 2003). In practice, precipitation has a negative correlation with atmospheric dust concentration, as discussed below. Although the temperature does not directly contribute to the dust flux equation in the CLM model, increasing the temperature leads to lower soil moisture (Seneviratne et al., 2010) and a higher possibility for dust emission. For more details about parametrizations and calculations, the readers are encouraged to see (Zender et al., 2003, and Oelsen et al., 2013)


From the model outputs, we derived all available columnar dust burden dataset (ranging from 0.058 µm to 3.65 µm) by the summation of the mean monthly values of the accumulation mode (particle size from 0.058 to 0.27 µm) and the coarse mode (particle size from 0.8 to 3.65 µm). The geographical focus of the current study's latitude and longitude is 15 °N to 45 °N and 20 °W to 62.5 °E (hereafter referred to as the MENA region). We further focus on the Middle East region (20

°N to 45 °N and 45 °E to 62.5 °E) since it has a higher population and is sensitive to changes in the dust events, as discussed in the introduction. Here, we use the regional and temporal averaged monthly surface temperature, near surface wind speed, precipitation, soil water, and total leaf area index between 2010 to 2097 to investigate the possible changes in the dust concentrations. The dust concentration is controlled by above-mentioned variables. Here, we conducted composite analysis between CTL, RCP8.5 and SAI simulations to identify the important factors that influence the temporal and

spatial changes of the dust concentration under the above-mentioned scenarios. In this study, all available ensemble members of the GLENS project are used to represent the present-day climate or CTL (2010-2029 period), the future climate or RCP8.5 (2078-2097 period), and the future climate under solar geoengineering or SAI (2078-2097 period) simulations. Table 2 represents the different simulations, acronyms, number of ensemble members, and period of the analyses used in this study. We also carried out the t-test with (99.9% confidence level) to determine whether the

differences between RCP8.5, SAI, and CTL simulations are significant. In all contour plots, regions with a confidence level of less than 99.9% are indicated with hatch lines (i.e., differences in these regions are not significant). In addition, for all scenarios, we calculate the monthly, seasonal, and annual averages of climate variables. Furthermore, we calculated the correlation coefficient of dust with other considered parameters for the Middle East region, with two important dust hot spots. The annual time series of all parameters for both RCP8.5 (2010-2097) and SAI (2020-2099) scenarios over the





Middle East region are used to calculate the correlation coefficients of atmospheric dust concentration with surface

temperature, near-surface wind speed, total leaf area index, precipitation, and soil moisture and are listed in the Table 3.

Additionally, the wavelet coherence is applied to provide an indication of times and the periods to show the temporal

correlation of dust with temperature, wind, precipitation, soil water, and leaf area index. In this study, the wavelet analysis

is carried out in two stages of the local wavelet coherence (LWTC) and global wavelet coherence (GWTC). LWTC, on

the whole, is a measure of the correlation between two signals in the time-frequency plane while GWTC provides the

cumulative WTC averaged over time. In addition to LWTC, we also analyse the GWTC as in some cases it is impossible

to separate signals occurring close to each other only through the LWTC. First, for each time series $(x_n, n = 1, ..., N$ with

uniform time steps $\delta t$, the continuous wavelet transform $(W_n^X(s))$ is defined as the convolution of $x_n$ with the scaled and

normalized wavelet (Grinsted et al., 2004):

$$W_n^X(s) = \sqrt{\frac{\delta t}{s}} \sum_{n'=1}^{N} x_{n'} \, \psi_0 \left[ (n' - n) \frac{\delta t}{s} \right], \psi_0(\eta) = \pi^{-1/4} e^{i\omega_0 \eta} e^{-0.5\eta^2}, \tag{2}$$

where $\psi_0$ is the Morlet wavelet, $\left[ (n' - n) \frac{\delta t}{s} \right]$ the complex conjugate, $s$ the wavelet scale, $\omega_0$ dimensionless frequency,

and $\eta$ dimensionless time. In this approach, to test the significance of the WTC, 1000 randomly constructed synthetic

series are generated by Monte Carlo methods (Grinsted et al., 2004). Then, the cross wavelet transforms $(W_n^{XY}(s))$ for

each pair of time series (i.e., $x_n$ and $y_n$) is defined as $W^x W^{y*}$ where * is complex conjugation. Next, the LWTC $(R_n^2(s))$

for each pair of time series is calculated as follows (Torrence and Webster 1998; Grinsted et al., 2004):

$$R_n^2(s) = \frac{|S(s^{-1} W_n^{XY}(s)|^2}{S\left(s^{-1} |W_n^X(s)|^2\right) S\left(s^{-1} |W_n^Y(s)|^2\right)}; \ S(W) = S_{scale}\left(S_{time}(W_n(s))\right), \tag{3}$$

where $S(W)$ is a smoothing operator, $S_{scale}$ smoothing along the wavelet scale axis, and $S_{time}$ smoothing in time. Finally,

we use the global wavelet coherence (GWTC) to identify timescales of relevance for the study. Periods of maximum time-

averaged power are characterized as the dominant timescale of variability. The time-averaged wavelet coherence is defined

as follows (Elsayed, 2006):

$$G_c(s) = \frac{|W^{XY}(s)|^2}{\left(\sum_{n=1}^{N} |W_n^X|^2\right)\left(\sum_{n=1}^{N} |W_n^Y|^2\right)} \tag{4}$$

This equation computes the correlation between two time series in the entire study period at the scale $s$. Statistical

significance of $G_c(s)$ is also calculated using Monte Carlo methods. First, red-noise time series of the same lengths and

autocorrelation coefficients as the two input time series are generated and then $G_c(s)$ is calculated for each pair of red-

noise series. The $G_c(s)$ distribution obtained for each scale is then employed to compute the significance of the global

coherence (Schulte et al., 2016).



## 3 Results

Figure 1a shows the climatology of the dust concentration over the MENA region in the control simulation and Fig. 1b indicates the population density map over the MENA region (available from the Socioeconomic Data and Applications

Center (SEDECA) a data center in the National Aeronautics and Space Administration (NASA) (https://sedac.ciesin.columbia.edu/). The climatology of columnar dust mass concentration over the MENA region is derived from all twenty ensembles for the control simulation from 2010 to 2029 (Fig. 1a). The climatology derived from Fig 1a suggests that there are five sub-regions in the MENA region showing the highest dust concentrations: Northwest Africa (R1), North Africa (R2), Northeast Africa (R3), Southwest of the Iranian plateau (R4) and Northeast of Iranian

plateau (R5). Therefore, we first attempt to explore the correlations between the dust mass concentration and precipitation, soil moisture in the top 10 cm, total leaf area index (TLAI), surface temperature, and near surface (10m) wind speed over the area of both whole MENA and whole middle east through the local and global wavelet coherence (LWTC and GWTC) analyses. In Fig. 2, in addition to the whole MENA region (left column), the results from LWTC and GWTC are also shown for the Middle East (right column) separately as this region encompasses two major hotspots (R4 and R5) of dust

emission and it has a denser population compared to North Africa (Fig 1a). In addition to LWTC, we also computed GWTC as in some cases LWTC cannot clearly separate the signals. As an example, in the soil moisture-dust LWTC for both the MENA and Middle East regions, it is impossible to separate two signals 0.5 (i.e., seasonal) and 0.75-1.5-year (i.e., annual) periods from each other, while the GWTC elucidate the separation of the two signals. Overall, the local and global coherences between the dust and variables of precipitation, soil moisture, leaf area index, surface temperature (ST),

and wind show strong correlation (>0.9) at three periodicities of seasonal (0.5-year), annual (0.75- 1.5-year), and multidecadal (22-year) (Fig. 2). Note that these periods are the dominant modes of variability in each considered variable that are important for understanding the event synchronization.

As can be seen in Fig 2, the strong correlations between the dust concentration and precipitation, soil moisture and leaf area index are roughly out of phase (left-pointing arrows in Fig 2) at seasonal (0.5-year) periodicities. Based on the DEAD

model, the possibility of dust emission is highly decreased by increasing the soil moisture and vegetation cover (Zender et al., 2003; Oelsen et al., 2013). On the other hand, the strong correlation between the dust and temperature is approximately in-phase (right-pointing arrows) at interannual (0.5- and 0.75-1.5-year periodicities) periodicities, that is, the dust concentration increases with an increase in temperature. At interannual scales, there is a lead-lag relationship between the wind and dust, since the black arrows in Fig. 2 are roughly down-pointing. Therefore, an increase in the dust

concentration takes place with a lag time relative to the near-surface wind. All the above strong correlations (in-phase, out of phase, or down-pointing) at interannual scales would be strengthened after 2040. At the seasonal period of ~0.5 years, the results for the Middle East are approximately the same as those for the whole MENA region, while at around annual periodicity (0.75 - 1.5 year), there are some discrepancies between the whole MENA and the Middle East, particularly for the near-surface wind.



At the seasonal/annual periodicities (<0.5- and 0.75-1.5-year), although the state of the correlation (positive or negative) between the dust and other variables is similar in both the MENA and the Middle East regions, there are differences in the patterns where they are broader and more continuous in the MENA region compared to those in the Middle East. While for the Middle East, there are some seasonal/annual patterns at the <0.5- and 0.75-1.5-year periods, which are even extended to 1-month periods. The possible reason is that in the Middle East, each hydrological year (the period between

October 1st of a given year and September 30th of the next) contains two seasons of wet and dry, and the majority of the annual precipitation falls during the wet season. October to December of a given year and January to May of the next are considered to be the wet seasons in the Middle East, particularly over Iran and Iraq. Additionally, the precipitation over Middle Africa, particularly at the lower latitudes that are considered here, occurs during the dry season of the Middle East. Note that over the MENA as a whole, particularly the Middle East, the high dust concentrations occur over June to

September, consistent with the dry season in the Middle East. Therefore, the presence of the seasonal correlation patterns between dust and other variables at the LWTCs for the Middle East is expected. Moreover, in the MENA as a whole, the patterns around the 0.5-year scales are almost continuous due to a large amount of monsoon precipitation that falls over the lower latitudes of the MENA region during summer (coincided with the dry season of the Middle East) which increases the average value of the precipitation across the whole MENA region. In fact, the southernmost part of North Africa (i.e.,

the southern part of the MENA region) receives its precipitation by the intertropical convergence zone (ITCZ), which is different than the Middle East precipitation sources (Nicholson, 2009; Schneider et al., 2014). The moisture that turns into precipitation over the Middle East during the cold and wet season is predominantly originated from the Mediterranean, the continental polar (also known as the Siberian), the continental tropical (also known as the Sudanian), and the Maritime polar air masses (Heydarizad et al., 2021).

At the multidecadal periodicities (>16-yr period in Fig. 2), for both the whole MENA and the Middle East regions, the correlation patterns are approximately identical, signifying that the long-term memory of the variables over the area from North Africa to the Middle East is controlled by the same sources. These sources are most probably the large-scale ocean-atmospheric circulations related to the Atlantic and Pacific oceans such as ENSO (El Niño–Southern Oscillation), PDO (Pacific Decadal Oscillation), NAO (North Atlantic Oscillation), and AMO (Atlantic Multidecadal Oscillation) (Notaro

et al., 2015; Ahmadi et al., 2019; Alizadeh-Choobari and Adibi, 2019; Rezaei and Gurdak, 2020; Rezaei, 2021) which indicates that these processes are realistically represented in the GLENS simulations consistent with the observations. The multidecadal periodicities in the precipitation and temperature variables over Iran in the Middle East have been demonstrated to be more affected by the low-frequency oscillations of PDO and AMO, respectively (Rezaei and Gurdak, 2020; Rezaei, 2021). There are strong in-phase correlations between the dust and temperature and near-surface wind speed

where right-pointing arrows are observed in their LWTC, signifying that the dust concentration over the region increases with the temperature and 10m wind speed. In contrast, the dust has a strong out of phase correlation with precipitation, soil moisture, and leaf area index. Given the above evidence, we conclude that the dust mass concentration generally decreases with an increase in the precipitation, soil moisture, and leaf area index, and a decrease in temperature and 10m





wind speed over the MENA region, particularly across the Middle East. Furthermore, the results of the correlation

coefficient of dust mass concentration and considered parameters over the Middle-East region indicated that the near-surface wind speed, leaf area index, and soil moisture have the most impact on reducing the dust concentration respectively for SAI scenarios. However, for the RCP8.5 scenario, the leaf area index, near-surface wind speed, and precipitation affect the reduction trend of dust concentration, respectively as listed in Table 3.

Fig. 3a shows the monthly mean values of the dust concentration across the MENA region for all scenarios. The dust

reduction in the MENA region for both the SAI and RCP8.5 scenarios (compared to the CTL simulation) is stronger during the spring and summer seasons (Fig. 3a). Figure 3b presents the time series of the regional annual mean values of the dust concentrations. This figure displays that dust mass concentrations tend to decrease in MENA under both the global warming and SAI scenarios by the end of this century. Fig. 3c-q show seasonal and annual changes of dust mass concentration mean value in the MENA region under different climate scenarios. The dashed line boxes show dust hotspots

in the current climate and the regions without hatch line indicates the regions where the changes are more than 99.9% significant level based on the student's t-test analysis. The reduction for the SAI scenario is generally larger than that for the RCP8.5 (Fig. 3b and q), although a strong reduction is found in each dry season under the RCP8.5 scenario (Fig. 3f and 3i).

The detailed analysis suggests that the maximum reduction of the dust concentrations in the MENA region (in both the

global warming and SAI scenarios) mostly results from the weakening of the dust concentration in the Middle East, rather than from the North Africa (Fig. 3f, g, I and j). Figures 4a and 4b show that the zonal and meridional mean annual dust concentration for the CTL, RCP8.5, and SAI scenarios, which are averaged over the whole MENA region, respectively. Figure 4c illustrates the climatology of the dust concentrations for the RCP8.5 scenario. The seasonal mean values of dust concentration under both the SAI and RCP8.5 scenarios are separately shown for the MENA and Middle-East regions

(Fig 4d and 4e). Overall, in Fig. 4a, the highest dust concentrations (up to 37 $\mu g/m^3$) are found across Northeastern Africa (i.e., 30-32 °E) and Middle East (i.e., 48-62 °E) while in Fig. 4b, the lower latitudes of 15-20 °N (i.e., Northern Africa) have the highest dust (up to 30 $\mu g/m^3$). Notably, these high dust concentrations coincide with the five major dust hotspots of R1 to R5 (Fig. 4c) where among them, R5 is the largest and strongest. Figure 4c shows only a portion of the R5 region while the R5 in combination with R4 constitutes a major dust source for the Middle East. A strong reduction is found in

the dust concentration across the Middle East (Fig. 4a) under both SAI and RCP8.5 scenarios relative to the CTL simulation, consistent with patterns shown in Fig. 3. Both Figures 4d and 4e illustrate that summer and, to a lesser extent, spring have higher dust concentrations than autumn and winter seasons under both the SAI and RCP8.5 scenarios across the whole MENA and the Middle East. Figure 4e further shows a significant reduction in the dust concentration for the dry season under both RCP8.5 and SAI scenarios across the Middle East and this reduction is more significant for the

RCP8.5 scenario after 2060.

In the following, we investigate changes in the different components that contribute to the changes in dust with regard to the different scenarios. The annual mean temperature responses to the different scenarios are shown in Fig 5a-c. As





expected, in the whole MENA region, the temperature would significantly increase by 3-6.5 °C under the high emission scenario (RCP8.5), while under the SAI scenario, it would slightly decrease by 0.5-1°C (Fig. 5), as also shown in (Kravitz

et al., 2017; Tilmes et al., 2018; MacMartin et al., 2019). Furthermore, the annual mean surface temperature for RCP8.5 and SAI scenarios are shown in Fig. 5d from 2010 to 2097. RCP8.5 shows strong temperature increasing, while SAI successfully maintain the average temperature as the level of CTL.

Figure 6 shows the spatiotemporal anomalies of TLAI for the different scenarios of CTL, RCP8.5, and SAI. Although the TLAI under the RCP8.5 scenario shows some significant reduction compared to the CTL across the whole region, except

the region between the Mediterranean and Caspian Seas (Fig. 6a, d, g and j). The monthly mean TLAI over the whole MENA region (Fig. 6p), shows that the TLAI slightly increases during the winter and spring seasons (i.e., mostly wet seasons). This increase in the monthly mean TLAI, despite the decrease in TLAI over the large geographical coverage of the MENA region, reveals that the averaged-TLAI is determined by the values from the Northern MENA region. Over the summer and autumn, there are no significant changes in the mean value of TLAI under the RCP8.5 scenario (Fig. 6p). On

the contrary, under the SAI scenario compared to the CTL, the TLAI shows a significant increase both spatially and temporally (Fig. 6b, e, h, k, p and q). The mean annual TLAI time series (Fig. 6q) also confirm these results where TLAI has a positive trend under the SAI scenario while it has no significant variations under the RCP8.5. Figure 6r reveals the annual number of the grid cell in the studied region with a TLAI of larger than 0.3 for both SAI and RCP8.5 scenarios. Although there is no considerable change in TLAI under RCP8.5, Figure. 6r indicates the significant increase in TLAI

under SAI simulations.

Figure 7 shows the 10-meter wind speed responses to different scenarios. In general, the wind speed tends to decrease under the global warming (RCP8.5) scenario compared to CTL across the whole region. Despite this reduction across the whole region, during summer there is an increase in the wind speed (up to 20%) under the global warming compared to the CTL across 15-30°N, containing the two major dust hotspots of R1 and R3 (Fig. 7g). SAI also shows reductions in

the wind speed compared to the CTL during all the seasons, and in the Middle East with two major dust hotspots, the reduction is even stronger (Fig. 7b, e, h, k, n, p and q). Figure 7q further shows that the wind speed for both the SAI and RCP8.5 scenarios are reduced, and under SAI, the wind speed reduction is gradually stronger than RCP8.5 starting from 2050. Figures 7p show that the decrease in the wind speed under the SAI scenario is larger than that from the global warming scenario. Figure 7r shows the annual mean of the near surface wind for all scenarios. It is evident that the regions

with higher latitudes (>32ºN) are indicated by a reduction in their wind speed (under both RCP8.5 and SAI compared to CTL).

The temporal and spatial differences in the top 10 cm soil water for different scenarios are shown in Fig. 8. In general, Fig. 8a, d, g, j, and m depict an increase in the soil water over the North Africa and a decrease across the Middle East under the RCP8.5 scenario (compared to the CTL). Notably, the absolute changes over the North Africa are larger

compared to the Middle East. On the contrary, under the SAI simulation (compared to RCP8.5), the soil water demonstrates a decrease over the North Africa and an increase across the Middle East (particularly during the spring





season). Figures 8p and 8q show the monthly and annual mean values of the soil moisture for different scenarios. In the wet seasons (winter and spring), the soil moisture for the RCP8.5 is lower than the CTL, while the SAI simulation shows higher values of soil moisture compared to the CTL (Fig. 8p). Furthermore, Fig 8q shows that a moderate positive trend

of the annual mean value exists in the soil moisture under the SAI scenario.

Figure 9 displays the spatiotemporal differences between the CTL precipitation and those obtained from the RCP8.5 and SAI scenarios. The results suggest that under the RCP8.5 scenario compared to CTL, the precipitation increases across the North Africa, especially in summer and fall seasons (Fig 9g and j). Also, under SAI simulation compared to the CTL, the Middle East receives higher precipitation especially in spring seasons (Fig. 9e). Comparison of the precipitation

changes between SAI and RCP8.5 simulations reveals that during the spring season, there is an enhancement in the precipitation under the SAI simulation compared to the RCP8.5 in the Middle East region (Fig. 9f). The monthly mean values of the precipitation from different scenarios (Fig. 9p) show that under the RCP8.5 scenario (compared to the CTL), precipitation is projected to significantly increase during the summer season, and under the SAI scenario (compared to the CTL), this region would experience more precipitation during the spring and summer. The time series of the annual mean

precipitation with its minimum and maximum (indicated by the error bars) is presented in Fig. 9q. It suggests that the mean annual precipitation in the MENA region does not significantly change under the SAI and RCP8.5 scenarios and the changes in precipitation are generally negligible across the whole region. On the contrary, Fig. 9r shows that the increased precipitation rate for the RCP8.5 and SAI scenarios is higher compared to CTL over the Middle East.

In summary in, Figure 10 shows the monthly mean values for the parameters under different scenarios across both the

MENA region (left column) and the Middle East (right column). The future dust concentration tends to decrease across both the MENA region (a) and the Middle-East region (b) almost equally for both RCP8.5 and SAI compared to CTL. Different factors may be leading to the decrease in dust. The increase in surface temperature and in wind speed in summer for RCP8.5, would lead to an increase in dust compared to the CTL and the SAI scenario. However, this may be countered by the increase in precipitation in summer in RCP8.5. On the other hand, the increase in TLAI and soil moisture for the

SAI scenario is expected to lead to a reduction in dust. The increase in both TLAI and the decrease in wind speed in winter and spring in both SAI and RCP8.5 may be for the most part responsible for the reduction in dust in the following season. reasonably consistent with an increase in the leaf area index (e and f), precipitation (g and h) and soil water (i and j), as well as a decrease in wind speed (k and l). Remarkably, these agreements and correlations are more obvious for the Middle East.

## 4 Discussion


Regarding the CTL simulation, the regions that are highlighted with the dashed-lined boxes in Fig 1a (i.e., R1, R2, R3, R4 and R5) are introduced as hotspots of the columnar dust concentration over the MENA region, and this is in agreement with the global-scale atmospheric dust sources determined by previous studies (e.g., Prospero et al, 2002; Ginoux et al.,





2012; Middleton 2017). The Saharan desert as the largest warm desert in the world encompasses R1, R2 and R3. Notably,

R3 is consistent with the Bodélé Depression in Northern Chad, as the region of highest dust concentrations in the world (Giles et al., 2005). Region R4 also covers some part of Iraq and Iran and this region accounts for one of the important sources of dust emissions in the Middle East region (Prospero et al., 2002; WMO and UNEP, 2013; Cao et al., 2015). Finally, the Central Asia, and the Karakum and Kyzylkum deserts are the main sources of dust storm generation (Orlovsky et al., 2005), which corresponds to region R5 in Fig 1a. In general, it is found that the locations and concentrations in the

dust hotspots regions are realistically simulated by the GLENS. The changes in the dust concentration over the Middle East might be considered even more important than those in the Northern Africa due to its higher population; although the future patterns of the population density may also change. Also, it is concluded that dust activities are more important in the area of interest during summertime with drier and warmer conditions (Fig 3 and 4).

The results from the wavelet analysis demonstrate that the dust concentrations in the considered regions are substantially

influenced by the changes in temperature, precipitation, near-surface wind, soil moisture, and leaf area index at both the seasonal/annual (0.5 and 0.75 - 1.5-year periodicities) and multidecadal (>20-year periodicity) scales (Fig 2). The results indicate that the dust concentration decreases with a) an increase in the precipitation, soil moisture and leaf area index and b) a decrease in the temperature and near-surface wind speed over the whole MENA region.

As our analysis reveals, the reduction of the future dust mass concentration over the MENA region (in both of the RCP8.5

and SAI scenarios) are mostly due to the weakening of the Middle East hotspots (Fig. 3 and 4). Moreover, the highest dust concentration of each year occurs over the Middle East during summertime (Fig. 4e) compared to CTL. The reduction rate of the dust concentration is about 5-40% for the RCP8.5 scenario (compared to CTL), where it is stronger from March to September period, especially for the dust emission in the Middle-East region (Fig. 3a, f, i, and l). Similarly, the dust concentration is also found to decline under the SAI scenario compared to CTL (Fig. 3d, g, j, and m) over the whole

MENA region. Dust concentrations in the summer of the Middle East and Northeast Africa (i.e., R3, R4, and R5) under the SAI scenario are approximately 10-30% higher than in the RCP8.5 scenario (Fig. 3k).

As depicted in the result section, the increase in the mean monthly TLAI for the RCP8.5 and SAI scenarios (compared to the CTL), is mostly determined by the values of the quantity in the Northern MENA region. This increase is probably because of $CO_2$ fertilization, the Northern MENA are covered with vegetation, and higher $CO_2$ in RCP8.5 and SAI boosts

plant growth (Ueyama et al., 2020). Fig. 6p demonstrates that the TLAI from the SAI scenario has increased up to ~0.2 (i.e., about 30%) compared to the CTL. In the spatial maps, this increase is more significant across the Northern MENA (i.e., southern Europe) with higher annual precipitation (i.e., tropical climate), which contains no dust hotspot. In the Community Land Model, 0.3 has been considered as a threshold of Leaf Area Index (LAI) for the dust emission, and for a region with LAI of less than 0.3, dust emission may be emitted (Mahowald et al., 1999; Zander et al., 2003; Mahowald

et al., 2010; Kok et al., 2014). Overall, the total leaf area index is found to increase under both the RCP8.5 and SAI scenarios whereas the increase under SAI is higher compared to the RCP8.5 scenario. Under RCP8.5, extreme heat and potential extreme drought will prohibit the TLAI from increasing (compared to the SAI), but under SAI, lower temperature





will benefit plants, and also reduce the latent heat which will increase soil moisture. While this increase in the TLAI is found to be small over the dust hotspots, this small increase may help to decrease the dust concentrations, since based on
wavelet analysis and correlation coefficient, there is a negative correlation between the dust and TLAI in the MENA region.

Our results show that the wind speed is generally weaker under SAI simulation compared to RCP8.5 throughout the year (Fig 7c, f, I, l, o, p, and q). Figure 7r demonstrates that the wind speed at the higher latitudes considered here (>32 °N), such as the Middle East (containing two major dust hotspots), would decreases to a larger degree under both the SAI and
RCP8.5 scenarios. This is the possible reason for the larger reduction in the dust concentrations over the Middle East compared to North Africa under the SAI and RCP8.5 scenarios compared to the CTL (Fig 3 and 4). Such wind change under different climate change scenario is expected to affect the subseasonal variability and circulation (Zagar et al., 2020). The mean monthly soil water series (Fig. 8p) demonstrate that the soil moisture for the three scenarios of CTL, SAI, and RCP8.5 is approximately the same during the summer and autumn. On the other hand, over the winter and spring, when
precipitation is significant, the soil moisture under the SAI scenario is higher than those for both the CTL and RCP8.5 scenarios. This is a possible reason for the lower dust concentration under the SAI scenario compared to the RCP8.5 scenario in the MENA region. Moreover, under RCP8.5 scenario (compared to the CTL) in the North Africa where contains three hotspots are located (R1 to R3), higher content of soil water in the vicinity of dust hotspots in spring, summer and fall seasons might explain the lower dust concentration.

It is also worthwhile to mention that under SAI simulation (compared to the CTL), two dust hotspots of R4 and R5 across the Middle East will encounter an enhancement in the precipitation, consistent with an increase in the LAI index. Also, for the SAI scenario a negligible change of precipitation in the dust hotspots are found in the North African (Fig 9b, e, h, k and n). The comparison of the precipitation changes between the SAI and RCP8.5 simulations reveals that during the winter and spring seasons, there is an enhancement in the precipitation under the SAI simulation compared to RCP8.5 in
the vicinity of the dust hotspots in the Iranian plateau (Fig 9c and f), while there is a reduction during summer and autumn seasons for the North African's dust hotspots (Fig 9i and l). Figure 9m, n, and r show that the Middle East region, with a semiarid climate would undergo more precipitation under the SAI scenario (compared to the CTL), which is an important factor on how the dust concentration is determined. Correspondingly, in the Middle East, the dust concentration under the RCP8.5 scenario will decrease at a larger rate compared to the SAI and CTL scenarios, respectively (Fig. 4e). For better
understanding, Fig. S1 presents the mean values of dust concentration, total leaf area index, soil water, and wind speed with a higher resolution for the Middle East. Detailed investigation of wind speed, soil water and dust concentration at summertime in the Middle East region demonstrate that a higher near surface wind speed in the R4 region and lower precipitation and soil water in the R4 region under the SAI simulations lead to more dust concentration compared to the RCP8.5 scenario (Fig 3k and Fig. S1).

Nonetheless, there are some limitations associated with the present work. First, the GLENS project uses only one model and we suggest future works consider other models to reduce the possible uncertainties associated with just using a single

one. Moreover, neither the current study nor the GLENS project suggests the SAI as an alternative way for the emission reductions and mitigation efforts. We also point out that the results presented in the current study should not be used as an indication for the real-world large-scale deployment of aerosols in the atmosphere.

**5 Conclusions**

This study projects the changes in atmospheric dust mass concentrations in the MENA region under the Stratospheric Aerosol Injection and high emission global warming (RCP8.5) scenarios compared to the current climate. Our results show that the future dust mass concentration would be reduced by 10-15% under both the RCP8.5 and SAI scenarios compared to the CTL; such reduction is slightly stronger for the SAI simulations compared to RCP8.5 over the whole
MENA region. Although for summertime, with more frequent dust events, a reduction is projected for the RCP8.5 scenario compared to SAI. The detailed analysis further indicates that in the summer and the vicinity of dust hot spots of northeast Africa (R3) and the Middle East (R4 and R5), more dust concentration is projected under the SAI scenario compared to RCP8.5. In other words, under the SAI simulations, the more densely populated Middle East area would encounter more dust events than the RCP8.5 scenario after 2060, but still, fewer view dust events compared to the present day. Although,
the dust concentration has the approximately same decreasing trend in all other seasons under both RCP8.5 and SAI scenarios in annual timescales for the MENA region and the Middle East. We further conclude that, over the coming 80 years, the dust mass concentration generally decreases with an increase in the precipitation, soil moisture, and leaf area index, and a decrease in temperature and 10m wind speed over the MENA region, particularly across the Middle East, and also the near surface wind speed and total leaf area index have the most impact on this reduction.

**Acknowledgments, Samples, and Data**

Khalil Karami and Seyed Vahid Mousavi are partially supported by The World Academy of Sciences (TWAS) (grant no: A-097- FR3240304784). The CESM project is supported by the National Science. The data from the GLENS simulations is publicly available via its website: http://www.cesm.ucar.edu/projects/community-projects/GLENS/ (DOI: 10.5065/D6JH3JXX).

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

**Table 1. Component of the ECSM1 used in GLENS project**

| Component | Version | Reference |
|---|---|---|
| Atmosphere | WACCM | Marsh et al. (2013), Mills et al. (2017) |
| Aerosol | MAM3 | Liu et al. (2012), Mills et al., 2016 |
| Land | CLM4.5 | Oleson et al. (2013) |





**Table 2. Number of ensembles for each simulation**

| Simulation | Acronym | Period | Number of Ensembles |
|---|---|---|---|
| current climate simulation | CTL | 2010 – 2029 | 20 |
| future climate simulation | RCP | 2078 – 2097 | 3 |
| feedback simulation | SAI | 2078 – 2097 | 20 |


**Table 3. Correlation coefficient between atmospheric dust concentration and different variables for RCP8.5 (2020-2097) and SAI (2020-2099) scenarios in the Middle-East region.**

| Correlation Coefficient | RCP8.5 Simulations | SAI Simulations |
|---|---|---|
| Dust & Surface Temperature | -0.86 | 0.71 |
| Dust & 10mWind Speed | 0.73 | 0.90 |
| Dust & Total Leaf Area Index | -0.82 | -0.88 |
| Dust & Precipitation | -0.57 | -0.70 |
| Dust & Soil moisture | -0.17 | -0.73 |



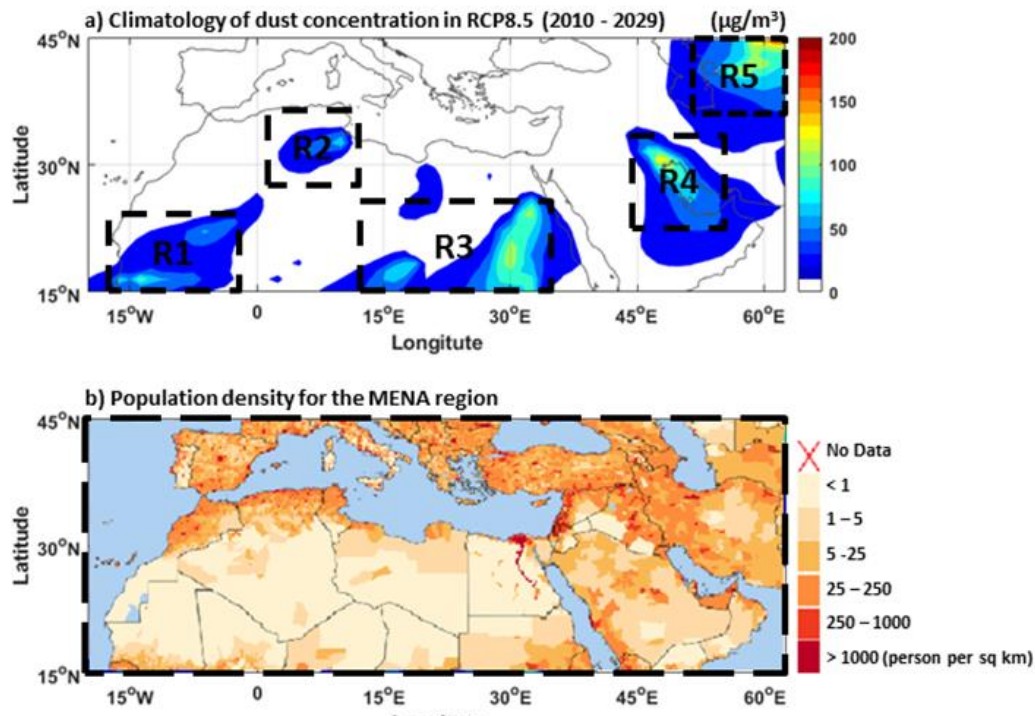

**Figure 1: a) The annual mean dust concentrations (µg/m3) over the MENA region over the period of 2010-2029 under RCP8.5. This is the average of twenty ensemble members. Dashed lined boxes (i.e., Northwest Africa (R1), North Africa (R2), Northeast Africa (R3), Southwest of the Iranian plateau (R4) and Northeast of Iranian plateau (R5)) show dust concentration hotspots, and b) Population density in the MENA region [(SEDECA: https://sedac.ciesin.columbia.edu/).].**



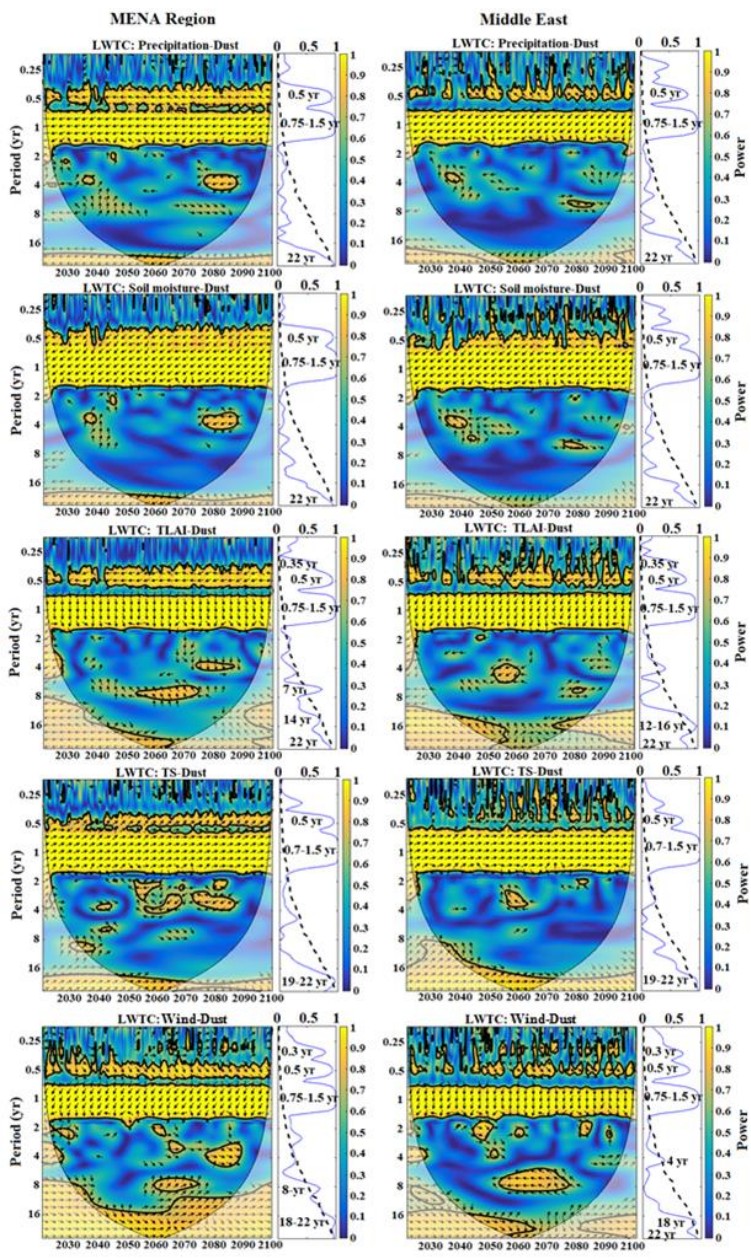

**Figure 2:** Local wavelet coherence (LWTC) and Global wavelet coherence (GWTC) in the SAI simulations between the dust concentration and precipitation, soil moisture, vegetation coverage (TLAI), surface temperature (ST), and wind speed over the whole MENA Region and Middle-East. Notably, the axis in the left-side is period (year) and the color-bar reveals the spectral
power for each LWTC that varies from zero to 1. The thick black curves indicate the 5% significance level, and the less intense colors denote the cone of influence (COI). Black arrows show the phase angle, characterizing the in-phase (right-pointing arrows) and anti-phase (left-pointing arrows) relations. Up- or down-pointing arrows are also indicative of lead-lag between the time series (Grinsted et al., 2004; Holman et al., 2011). In the right-hand side of each LWTC, its corresponding GWTC is shown through blue curve. In each LWTC, the dotted-black curve is the significant test (95% significant level). The dominant
modes (e.g., 8-yr) are also written on each GWTC curve.



**Figure 3: The monthly (a) and annual (b) mean values of the dust concentration for different scenarios, (c-q) Seasonal and annual changes of dust mass concentration mean value in the MENA region under different climate scenarios. All available ensemble members of the GLENS project are used to calculate mean value of dust concentration for CTL (2010-2029), RCP8.5 (2078-2097) and SAI (2078-2097). The dashed line boxes show dust hotspots and the regions without hatch line shows student's t-test analysis with 99.9% significance level.**

**Figure 4: Mean values over different latitude (a) and different longitude (b) of annual dust concentration in the MENA region of CTL (2010-2029), RCP8.5 and SAI (2010-2029) and (c) climatology of dust concentration of RCP8.5 (2010-2029) mean values are calculated for all available ensemble member in the GLENS dataset, dash-line in (c) shows Middle-east region. (d) The seasonal mean dust concentration in the MENA region under SAI and RCP8.5 simulations. (e) The seasonal mean dust concentration in the Middle-East region under SAI and RCP8.5 simulations**






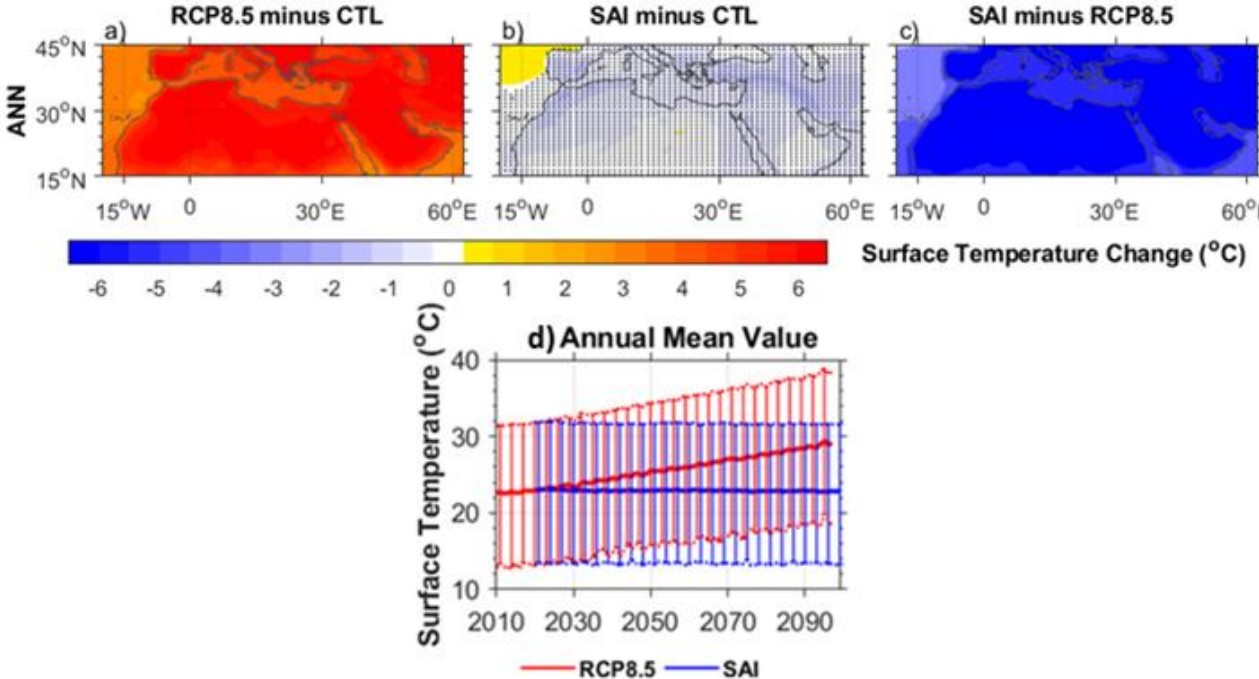


**Figure 5: Annual changes of surface temperature mean value in the MENA region under different climate scenarios (a - c).**
**All available ensemble members of the GLENS project are used to calculate mean value of surface temperature for CTL**
**(2010-2029), RCP8.5 (2078-2097) and SAI (2078-2097). (d) The annual mean values of the dust concentration for different**
**scenarios are shown, error bar in (d) show maximum and minimum values of annual surface temperature.**




**Figure 6: The same as Fig 3 but for the total leaf area index (TLAI) differences (TLAI is a unitless parameter). Error bars in Fig. 6q also show the maximum and minimum values of TLAI for different scenarios. (r): The time series of the annual mean total leaf area index higher than 0.3 (that is considered as minimum threshold for the dust emission) in the MENA region.**






**Figure 7: The same as Fig 3 but for the wind speed differences. r) is mean value of annual wind speed over different latitude for MENA region.**






**Figure 8: The same as Fig 3 but for the top 10 cm soil water.**

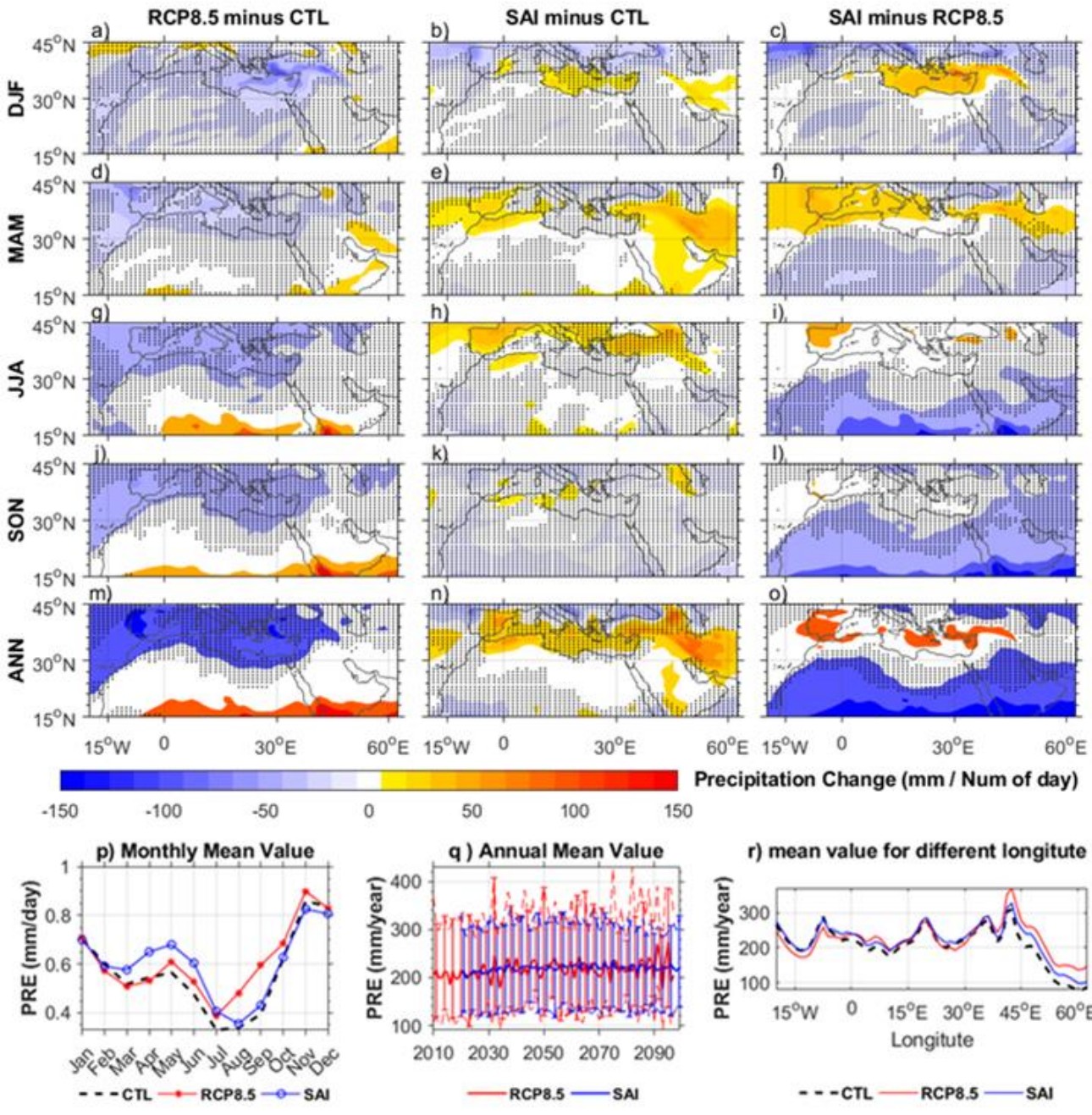

**Figure 9: The same as Fig 3 but for the precipitation. r) Is mean value of annual precipitation over different longitude for**
**MENA region.**

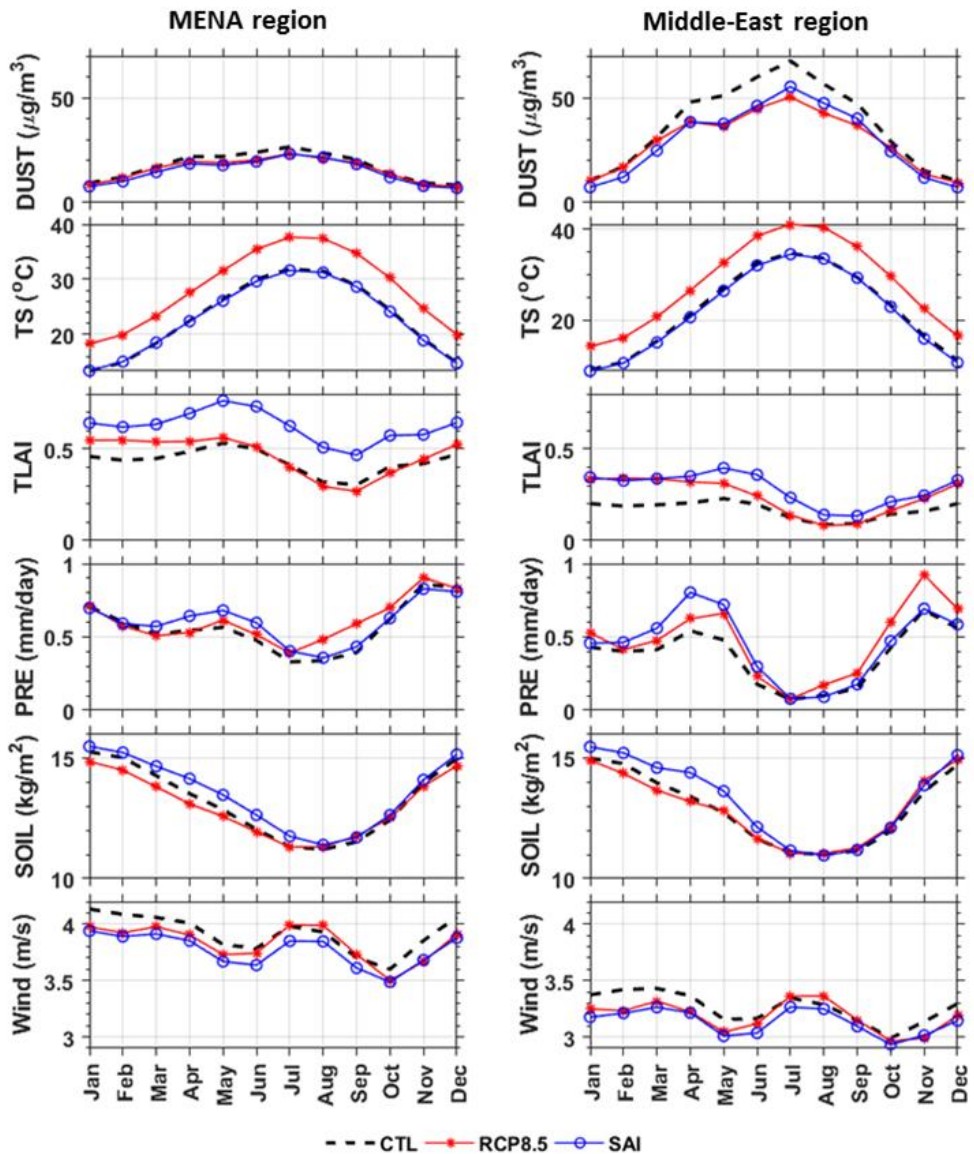

**Figure 10: The monthly mean values of the five variables that are important in deriving the future changes of the dust concentrations in the MENA region (left column) and the Middle East region (right column).**
