# Peer review of "Future dust concentration over the Middle East and North Africa region under global warming and stratospheric aerosol intervention scenarios"

_Atmospheric Chemistry and Physics, 2022_

## Author Response (AR1)

**To**: Editor, ACP

**Subject**: Author Comments of Manuscript, acp-2022-270

**Dear Dr. Jianping Huang,**

Upon the recommendation, we have carefully revised the manuscript entitled "*Future dust concentration over the Middle East and North Africa region under global warming and stratospheric aerosol intervention scenarios*" after considering all the comments and suggestions made by the Referees; all the changes made in the revised manuscript are highlighted in yellow. The following is the point-to-point response to all the comments (the reviewers' comments are rewritten in black color and the replies in blue). We appreciate the opportunity to revise our paper. We believe that the manuscript is much improved after positively addressing all the requested revisions. In the following we provide answer of Anonymous Referee #1, and Anonymous Referee #2 respectively. The main changes have been made in the new version based on the referee's comments/suggestions are as follows:

- We replaced Fig.2 with two new figures, a new figure (Fig. 9) for detailed analysis of the correlation between dust and considered variables, and a second figure (see next point).
- We provide a new figure for annual trends (Fig. 10) of all considered variables over dust hot spots to interpret the positive and negative correlation considering ascending or descending trends
- New Table for the correlation coefficient over dust hot spots, shows which variable would have more effect on the change of future dust concentration in different regions
- New figures for monthly trends using box plots (Fig. 11 and Fig. S1), to give a better view of the statistical analysis of different scenarios
- Rewrite the result section with three subsections to increase readability
- Rewrite the result, discussion, and conclusion sections based on new findings and figures
- To magnify the parameter's changes over the dust hotspot regions, these regions are specified by dashed lines, over all contour plots.

**Notice:** The line and page numbers refer to the pdf file of "Revised Manuscript".

**Response to the Anonymous Referee #1:**

Major concerns:

a) The cross-coherence analysis:

I don't find that the method is very well explained in section 2. What are the connections to the axes in Fig. 2? How do you come from the equations to the quantities (amplitude, phase?) shown in the figure? More importantly, I am also confused about the physical interpretation. There are probably annual cycles in all meteorological variables. This means that there will always be coherence between them. As the annual cycle probably is different from a pure sinusoidal, there will also be a signal at 1/2-year. So what do we actually learn from Fig. 2? In the discussion section (l365) it says that the dust is 'substantially influenced' by the changes in the other fields. But I don't think you can conclude that from the analysis. What we learn is only that there is an annual cycle in all the fields including the dust but nothing about the physical interpretation. This manuscript presents projected changes of dust storms in the Middle East and North Africa region, which is very sensitive to climate change, under future scenarios, with (SAI) and without (RCP8.5) geoengineering, focusing on the stratospheric aerosol intervention effect but discussing both of them. Therefore, the title could be modified to reflect better the content.

Reply: We thank the reviewers for the comments and suggestions. We think that by implementing the reviewers' comments and suggestions, the revised version has significantly improved.

As you mentioned correctly, Fig 2, mostly demonstrates the annual correlations between dust and considered variables. In new version of the manuscript, Fig. 2, has been replaced by two new plots (Fig. 9 and 10) to make correlations more understandable. In Fig. 9, we depicted the correlation coefficient of dust with considered parameters for all grids (i.e., cells with a horizontal resolution of 0.9° latitude by 1.25° longitude) over the MEAN region. The positive and negative correlations are shown as a contour plot for both RCP8.5 (left column) and SAI scenarios (right column). Moreover, the dust hotspots are specified using the climatology of dust concentrations in this figure (regions with no hatch-lines). The correlation coefficients for hotspots are listed in Table 3. Furthermore, annual trends of parameters over the dust hotspots are shown in Fig. 10, for both RCP8.5 and SAI scenarios. This figure depicted the intensity of annual changes for any parameter in any special region that would affect the dust change.

Regarding your suggestion about the title, as you mentioned MENA region is very sensitive to any changes in future dust. Although our focus in this study is on the stratospheric aerosol intervention geoengineering effect, in this paper we would like to give a perspective of future dust under both RCP8.5 and SAI scenarios, and we selected a title to reflect this issue (i.e., changes of future dust under both RCP8.5 and SAI scenarios). Moreover, the sections of the

manuscript have been rewriting based on the new plots. The following lines and Figure were added to the new version of manuscript:

[revised manuscript text omitted]

b) The rest of the paper seems to me to be too much focusing on presenting the details about the changes in the different fields. I think many of the panels basically shows the same and that the number of plots and panels could be reduced. I really miss some solid analysis and results about what drives the changes in the dust. The dust generally decreases in the RCP8.5 scenario but it decreases further in the geoengineering scenario. Perhaps I am missing something but I could not find an explanation. The correlations in Table 3 could be a beginning, but the physical connection between the variables requires that the trends - which I guess determines most of the correlations here - are removed.

Reply: as you suggested, plots with the same content have been removed. Fig. 1 to 4 in the old version are combined and depicted as Fig. 1 to 3 in the new version. Fig 10, containing several same panels for with other figures for the monthly mean values, has been replaced by a new one for different hotspot regions (Fig. 11 for R4 and R5 dust hotspots and Fig. S1 for R1 to R3 dust hotspots in the new version).

As Fig. 2 and Fig. 3m (of the manuscript) depicted, the projected dust concentration has more reduction over dust hotspots for RCP8.5 scenario, compared with SAI scenario (Fig 3d, e, g and h, in new version), although, over the whole MENA region more reduction for SAI scenario has modelled (Fig. 2a-b).

As mentioned in the manuscript, because the dust emissions physically reduce with cooler temperatures, weaker winds, and wetter climate through increasing precipitation and soil moisture and, in turn, denser and broader vegetation coverage, all investigated parameters could affect dust concentration changes. Moreover, regarding the extent of the MENA region, different parts of this region have a special hydro-climate cycle, and subsequently, the different factors could be counted as a driving factor of the atmospheric dust change in different regions. By focusing on the dust hotspots, we find that, under both RCP8.5 and SAI scenarios the reduction of future dust is mainly controlled by the weaker surface wind, except for the R4 region and under the SAI scenario where an enhancement of the precipitation more than other parameters effects on the dust reduction. In the revised manuscript, Table 3 is replaced with new one with more detail about the correlation coefficient of each variable in the five dust hotspot regions. Moreover, the new figure of annual trends for parameters over the dust hotspots help the reader to understand the physical connections between the changes in dust concentrations and considered parameters (please see Fig. 10 in the new version).

Minor comments:
l54: reginal -> regional
Reply: done

l97: So this is an ensemble based on a single climate model? How are the different ensemble members generated?
Reply: the following sentences to the manuscript;
Line 97-100: For each ensemble member, the atmospheric state is initialized with 1 January conditions taken from different years between 2008 and 2012 of the reference simulation and a round-off (order of $10^{-14}$ K) air temperature perturbation, while the land, sea ice, and ocean start from the same initial conditions for each ensemble member.

l103: What is 'interhemispheric temperature gradient'?

Reply: the following sentences to the manuscript;

Line 104-107: The interhemispheric surface temperature gradient is defined in equation (1) of Kravitz et al (2017). It is simply the difference between the mean surface temperature in the Northern and Southern hemispheres. In the study of Tilmes et al (2018), the values for the interhemispheric differences for the different periods and scenarios are presented in Table 3 (T1).

l115-130: Is this a new method adopted for the present study? Is it described in the literature before? If it is new perhaps it should be described in more details and more background given. As it is now it is not transparent for me. For example, what is a transport bin?

Reply: this is not a new method, and the Dust Entrainment and Deposition model (DEAD) is used for the atmospheric dust mobilization scheme. Interested readers could find the detail about model in the lines 118-142 of the manuscript, and for more information could refer to the references mentioned in the manuscript.

l148: composite analysis? Is this the right word? You calculate the difference of temporal means.

Reply: the "composite analysis" phrase is replaced by "calculated the differences".

l160, Table 3: Are the correlations averages over all the ensemble members? It should be mentioned in the caption that this is annual means.

Reply: as requested, we add the following sentences in the caption: "The correlation coefficients are calculated using annual mean time series resulting from the average of all ensemble members, and spatially averaged over the corresponding dust hotspot region".

As mentioned I have problems with the presentation of the wavelet coherence. In line 171 why is [(n'-n)dt/s] the complex conjugate? Is omega_0 a constant? If it is how is it selected?

Reply: as mentioned above, to increase the understandability of correlation we replace Fig.2 with two new figures, and all sentences related to wavelet analysis have been removed.

l172: The sentence 'In this approach .. ' seems misplaced here and should be moved down near line 184.

Reply: as mentioned above the sentences related to wavelet coherence are removed.

More importantly in Fig. 2 the coherence is shown as function of time (x-axes) and period (y-axes). It is not clear from the text what these correspond to in the formulas.

Reply: as mentioned above the sentences related to wavelet coherence are removed.

Furthermore, the figure caption mention both the power and the phase which is not described in the text. The same goes for the cone of influence.

Reply: as mentioned above the sentences related to wavelet coherence are removed.

Eq. 6: Should there not be some smoothing here too?

Reply: as mentioned above the sentences related to wavelet coherence are removed.

The discussion of Fig. 2, page 7-8:
It should be pointed out more specifically in the text that Fig. 2 is for SAI. Does it look the same for the RCP8.5? Why focus on the SAI here?

Reply: as mentioned above the sentences related to wavelet coherence are removed.

The 22-years variability and variability larger than 16 years seems to be outside the cone of influence. Also, it is not significant in the GWTC. In general, the two regions in Fig. 2 look identical to me. I don't think you can say that there are significant differences.

Reply: as mentioned above the sentences related to wavelet coherence are removed.

And I don't really see any change after 2040. Perhaps just presenting the GWTC would be better.

Reply: as mentioned above the sentences related to wavelet coherence are removed.

l207: 'Out of phase'. Does this mean -180? Is it just difference in sign?

Reply: as mentioned above the sentences related to wavelet coherence are removed.

l246: How does this indicate that the model is consistent with observations? There are no observations used in the present study.

Reply: as mentioned above the sentences related to wavelet coherence are removed.

Table 3: Why the big difference between RCP8.5 and SAI for temperature correlations? Is this table only discussed in l258?

Reply: According to the time series, of surface temperature and dust concentration, these results for correlation are anticipated. Based on the model's output and as expected, the surface temperature has no considerable change under the SAI scenario. In contrast, the MENA region's surface temperature rises up to 6 degrees for the RCP8.5 scenario. In the meantime, a reduction is projected for dust concentration under both scenarios. It seems that the increasing trend of annual mean surface temperature alongside the decreasing trend of the annual mean dust concentration is the reason of the negative correlation under RCP8.5 scenarios. The results of Table. 3, are discussed in line 267-271.

Section 3 should be split in two or more subsections. Perhaps not start with the coherence?.

Reply: As you requested, the result section divided in three subsections as below (see lines: 170-290 of new version of the manuscript)

3. Results

3.1. Atmospheric Dust Concentration change under Different Scenarios

3.2. Candidate Variables change under Different Scenarios

3.3. Correlation of Atmospheric Dust Concentrations with Candidate Variables

**Response to the Anonymous Referee #2:**

This paper is a straight forward comparison of predictions of model ensembles using one model with two scenarios, global warming (RCP8.5) and stratospheric aerosol injection (SAI), over 80 years compared with a control run of 20 years. The variable of interest is dust and its correlation with surface temperature, leaf area index, precipitation, soil moisture and wind speed. The region of interest is north Africa and the middle east with various dust hot spots identified. The bulk of the paper rests on describing Figs 3-9 which show the spatial and temporal variation of each of these parameters under the two scenarios for monthly and annual means. The spatial differences are shown variously and absolute value or percentage depending on the variable. It is not clear why they are not all shown as percentages.

Reply: We thank the reviewers for the comments and suggestions. We think that by implementing the reviewers' comments and suggestions, the revised version has significantly improved.
In this investigation plots of absolute changes and percentages of changes have been plotted for all investigated parameters. In addition, a plot with a better description for each parameter is included in the article. We agree that showing percentage change is helping to visualize both areas changes in dust hot spots. However, for regions with very small background concentrations, for example for Europe or regions with less than 5 ($\mu g/m^3$), and even a 50 or 100% change in dust, relative changes do not make sense (Fig. 1).

The authors then make some conclusions about the differences between the RCP8.5 and SAI scenarios, a number of which are difficult to believe if the error bars are included in the discussion of the annual differences or trends in for example soil moisture, wind speed.

Reply: considering your great suggestion, and for statistical analysis we added box plots (i.e., percentile values of standard deviation) to the monthly mean values. Furthermore, we depicted the standard deviation values of all available ensemble members (indicated by shaded envelope) in the annual mean value and other trends. The new plots including errors would give a better sense to readers of the statistics of the parameter changes under different scenarios. As mentioned in the text (previous version of the manuscript), the error bars plotted on the annual time series indicate the parameter's minimum and maximum value in that year. In the new version, the standard deviation (indicated by the shaded region) is used instead of the minimum and maximum.

[Figure]

**Figure 1: (a-o) Seasonal and annual changes of dust mass concentration mean value in the MENA region under different climate scenarios. All available ensemble members of the GLENS project are used to calculate mean value of dust concentration for CTL (2010-2029), RCP8.5 (2078-2097) and SAI (2078-2097). The regions without hatch line shows student's t-test analysis with 99.9% significance level.**

Error bars should be included on all the figures showing mean values: monthly, annually, or spatially. Currently error bars are included only on the annual means. The same should be to Fig. 10. Then the authors discussion of notable differences can be placed in the context of how well any one variable is known.

Reply: As mentioned above, the error bars have been considered for all of the monthly, annual, or spatial analyses using shaded region or box plots. Moreover, considering referee 1 comments and suggestions, we replaced Fig 10 (in the previous version) with two new figures Fig. 2 and Fig. 3 of the text (i.e., Fig 11 and S1 in the new version) and related sentences in the context are revised.

One of the results which is rather striking, but which the authors largely ignore, is how little difference there is between the various variables, except for surface temperature and leaf area index, for the two scenarios, see e.g., Fig. 10. Similarly for most variables there is primarily little difference between the two scenarios and the control. Isn't this surprising given one scenario is global warming as usual, whereas the other is to deal with global warming. Are we to conclude that only primarily temperature will be affected?

Reply: The little difference you correctly pointed out in Fig 10 (previous version), could be the result of spatially averaged over the large area of MENA and the Middle East. The new figures of multi-monthly mean values with error bars Fig. 2 and Fig. 3 of the context (i.e., Fig. 11 and S1), alongside the contour plots over dust hotspots shows considerable differences between different scenarios more clearly and shows that the change of dust concentration over the hotspots is influenced by changes in the surface wind speed, precipitation, and vegetation cover (Please see Fig. 2, 3, 5, 7, 11, and S1 in the manuscript).

Moreover, using the atmospheric dust mobilization scheme, the surface temperature does not participate directly (see equation 1), so, we investigated five parameters that can directly or indirectly contribute to dust events to find the most effective variable for decreasing the dust concentration in this region. By the end of the century, the average temperature remained constant at the 2020 level in the geoengineering scenario, and for the RCP8.5 scenario, approximately 6 degrees increase in temperature was projected for the studied area. At first glance, the increase in temperature causes lower soil moisture and, subsequently, more probability of the formation of dust event. While for both scenarios, despite the temperature remaining constant or increasing, dust reduction has been projected over the studied region.

The paper would be improved if some discussion along these lines was added and if the authors treated the supposed differences and trends more carefully to put them in the context of the uncertainty in the knowledge of variable in question. If differences or trends are small fractions of the uncertainty, there cannot be much confidence in such predications.

Reply: As you suggested the error bars are considered for all analysis. Moreover, to decrease the uncertainty in mean monthly and annual trends, we investigate these trends over dust hotspots instead of entire MENA and Middle-East regions.

More detail on these and other points follow in paper order, including a couple of minor points.
44 From remote "regions?"
Reply: this sentence means is "MENA cannot receive humidity transferred from other regions".

[Figure]

**Figure 2: The multi monthly mean values of the considered parameters with percentile values as error bars for R4 dust hotspot (left column) and R5 dust hotspot (right column), for different scenarios. The box plots are depicted with the median (horizontal line), the 25–75 percentile (box), the 5–95 percentile (horizontal line), and outlier data (circle).**

[Figure]

**Figure 3: The multi monthly mean values of the considered parameters with percentile values as error bars for R1, R2 and R3 dust hotspots, for different scenarios. The box plots are depicted with the median (horizontal line), the 25–75 percentile (box), the 5–95 percentile (horizontal line), and outlier data (circle).**

80 dioxide

Reply: is implemented.

166 Isn't it the cumulative LWTC averaged over time? Or is there a new variable WTC?
Reply: in response of Referee 1, and to show the correlation of dust concentration with considered variable we replace this figure (Fig. 2 of the previous version) with new figures.

Fig. 2 Some general comments should be made to explain the similarities of all the figures no matter the variable being correlated, particularly for readers not accustomed to such plots. For example, why is there always a strong annual cycle? Is this just the strong annual seasonal cycle? Why is there a definite semicircle traced out delineating the bright and dim colors in all plots? Is this an issue with the period versus the year, i.e., there can't be an eight year correlation for times less than 16 years beyond the start date? Presumably this is the cone of influence. But if that is the case why are there any correlations outside this cone shown on the figure?
Reply: as mentioned above the Fig. 2 and sentences related to wavelet coherence are removed.

Fig. 2 caption is unclear. 1) Isn't the cone of influence denoted by the more intense colors? If that isn't the case then it suggests the cone of influence is only from 2-20 years before 2050 and after 2070 with no influence in the center of the figure? 2) What is meant by the whole MENA region. Is that different than the MENA region? Also in the text line 199, and similarly confusing whole middle east. These regions were defined clearly earlier, now there seems to be a confusion about what they mean.
Reply: as mentioned above the Fig. 2 and sentences related to wavelet coherence are removed.

218 Again the whole MENA compared with the Middle East. Is this now not the whole Middle East?
Reply: as mentioned above the Fig. 2 and sentences related to wavelet coherence are removed.

Fig 3 c-q. Consider using percentages. The average reader may not know if 45 ug/m$^3$ is a lot or a little. But checking Figs 3a, b indicates that 45 ug/m$^3$ is 50-100% above or below the mean value, so it is a lot.

Reply: as mentioned and depicted above in the Fig. 1, if we use the percentage for the dust concentration, a decrease or increase of 50% or more are shown for some regions in Europe with no dust hot spots. For more explanation we added the following sentences to the discussion section;

lines 305-313: As our analysis reveals, the reduction of the future dust mass concentration over the MENA region (in both of the RCP8.5 and SAI scenarios) are mostly due to the weakening of the Middle East dust hotspots (Fig. 2 and 3). Moreover, the highest dust concentration of each year occurs over the Middle East during summertime (Fig. 2f and g). The reduction rate of the dust concentration is about 5-40% for the RCP8.5 scenario (compared to CTL), where it is stronger from March to September, especially for the dust emission in the Middle-East region (Fig. 2a, Fig. 3d, g, and j). Similarly, the dust concentration is also found to decline under the SAI scenario compared to CTL (Fig. 3b, e, h, and k) over the whole MENA region. Dust concentrations in the summer of the Middle East and Northeast Africa (i.e., R3, R4, and R5) under the SAI scenario are approximately 10-30% higher than in the RCP8.5 scenario (Fig. 3i).

Figs 6-9 q) which depict the annual mean value. Don't all of these figures, except fig. 6q) show that considering the error bars there is no difference between RCP8.5 and SAI. The difference in the means is a small fraction of the range of differences mapped out by the error bars. The differences shown in the monthly mean value figs p) appear at first more significant, but where are the error bars on this figure? If they were included the picture might be just as difficult in concluding a difference between RCP8.5 and SAI. Of these figures the only two that show a distinct difference outside the error bar range are surface temperature and TLAI.

Reply: As mentioned before, we considered error bars for all monthly and annual trends. As depicted in Fig. 2a, b and also Fig (5-8) p and q, the monthly and annual differences of the scenarios are seen. For example, Fig. 5p, q and r, clearly show that the TLAI increased significantly under the SAI scenario, and also the model projected an enhancement for the TLAI during winter and spring under RCP8.5. On the other hand, according to the algorithm implemented in the GLENS project, the considerable difference in temperature between the SAI and RCP8.5 scenarios is acceptable. Moreover, to reduce uncertainty we focused on the dust hotspots instead of the MENA region. The error bars on monthly mean values and annual trends in the Fig. 2-8, alongside the new figures (i.e., Fig 10, 11, and S1), depicted the difference between scenarios more clearly.

Thus the authors conclusions such as at lines 311-, "Figure 7q further shows … and under SAI, the wind speed reduction is gradually stronger than RCP8.5 starting from 2050.", or 324, "Fig 8q shows that a moderate positive trend of the annual mean value exists in the soil moisture under the SAI scenario." are deeply flawed. There is no trend that would stand under any statistical test given the size of the error bars on the data. The authors must be much more careful about what can be concluded from these monthly and annual mean values.

Reply: considering your great suggestion, and to investigate the statistics on monthly mean values, we include error bars on monthly variations plots, and tried to rewrite and update the

manuscripts regarding new plots with error bars (Fig. 11 and Fig. S1). These figures are discussed in on the manuscript on line: 282-290 as a below:

"Figure 11 included error bars for monthly mean values of all considered parameters for R4 and R5 regions, and shows considerable reduction of dust concentration between the control and the two future scenarios for both regions in spring to fall with the stronger differences for R5. Differences between RCP.85 and SAI are however not significant. The monthly mean values with error bars of all considered parameters for R1, R2 and R3 regions are also shown in Fig S1. The reduction of the monthly mean value of dust concentration over the R4 region (Fig. 11a) may be a result of the increase in precipitation (Fig. 11e) and soil moisture (Fig 11g) the decrease in wind speed (Fig. 11c). Moreover, it seems that the reduction of dust concentration over the R5 region (Fig. 11b) is mainly controlled by the lower wind speed (Fig. 11d) and higher leaf area index (Fig. 11j). The results of Fig. 10 and Fig.11, are in good agreement with the results and correlation coefficients in Table 3."

Similar comment can be made about Fig. 9r), a slight difference appears in the mean values east of 50 degrees, but would this appear significant if the error bars were included on this figure? The error bar range is on the order of plus/minus 100 mm/year.

Reply: We considered the statistical analysis for your mentioned figure for all available ensemble members and depicted in new version in the Fig. 7r. In the previous manuscript, the error bar in annual trends indicated the maximum and minimum of parameters and it replaced by the standard deviation of the annual mean values for different ensemble members in new version. The model simulates an annual mean of the precipitation almost 220 (mm/year) over the entire MENA region for the CTL scenario (Fig.7q). For longitudes > 40 °E (i.e., in the vicinity of R4 and R5), the differences between the RCP8.5, SAI, and CTL scenarios is about 20-50 mm/year (Fig. 7r). This means that mentioned region receives 10-25% more precipitation in the future climate and this is a considerable amount for this semiarid region.

Fig. 10. Error bars should be included on this figure, just as they have on all the annual means shown. This is needed to put the differences noted in the context of the overall uncertainty in the predictions.

Reply: As you suggested the error bars are included in the monthly mean values. Moreover, to decrease the uncertainty in mean monthly and annual trends, we investigate these trends over dust hotspots instead of vast MEAN and Middle-East regions. Please see Fig. 2 and 3 (Fig. 11 and S1 of the new version of the manuscript).

---

## Author Response (AR2)

**To**: Editor, ACP

**Subject**: Author Comments of Manuscript, acp-2022-270

**Dear Dr. Jianping Huang,**

Upon the recommendation, we have carefully revised the manuscript entitled "*Future dust concentration over the Middle East and North Africa region under global warming and stratospheric aerosol intervention scenarios*" after considering all the comments and suggestions made by the Referees; all the changes made in the revised manuscript are highlighted in yellow. The following is the point-to-point response to all the comments (the reviewers' comments are rewritten in black color and the replies in blue). We appreciate the opportunity to revise our paper. We believe that the manuscript is much improved after positively addressing all the requested revisions. In the following we provide answer of Anonymous Referee #1, and Anonymous Referee #2 respectively. The main changes have been made in the new version based on the referee's comments/suggestions are as follows:

- we have hatched significant regions instead of the insignificant regions in the contour plots (Fig. 3 to Fig. 8), to give better sense to reader.
- We used the detrended annual time series, for calculation of the correlation coefficient in Table3 and Fig. 9
- We performed statistical significance test on the annual time series to determine the confidence level of the correlations.
- we have added some sentences to explain the statistical analysis t-test in detail
- We have replaced all the contour plots (Fig. 3 to Fig. 8), that show the percentage of changes for different scenarios (instead of relative change in the previous version).
- We organized the rows of the Fig. 9, 10, 11, S1 and Table3, with the same order of parameters to increase readability.

**Notice:** The line and page numbers refer to the pdf file of "Revised Manuscript".

**Response to the Anonymous Referee #1:**

We thank the reviewers for the comments and suggestions. We think that by implementing the reviewers' comments and suggestions, the revised version has significantly improved.

**Major points:**

1) I am confused about the statistical significance and the way it is shown in the figures. In line 161 it is stated that the hatched areas are not significant. This choice is also mentioned in the caption to Fig. 3. But then in Fig. 3 there are areas with little change (shown with white) that are significant while some areas with large change (blue) don't seem significant. The same holds for the other figures except for Fig. 4 where it makes sense that the 'SAI - CTL' is not significant. In line 208 the differences in Fig. 5 are described as significant across the whole region while almost everything is hatched in that figure.

I also think the t-test should be described in more details. For example, what is the number of independent points used. And I think it would make more sense to hatch the significant regions instead of the insignificant regions.

Reply: Considering to the referee's suggestion, in the new version of the manuscript, the areas with significant difference have been hashed.

The white color in the contour plots indicate the areas in which the two scenarios have a small difference (less than 1% or 5% in the new figures). In other words, the color bar shows the differences between the means of the two scenarios. While the significance of the difference is defined by the t-value. and it depends on the means and variabilities of the two datasets (i.e., means, variances and the number of samples). Depending on the confidence level, the obtained t-value can be lower or higher than the statistical analysis threshold value. A brief explanation of the t-test statistical method has been added to the manuscript as a below

Lines 165-178: In this study, the independent t-test analysis has been used for comparing the statistical difference between scenarios for considered parameters. The t-test analysis is a statistical test that is used to determine the statistically significant difference of two samples. Depending on the confidence level, the obtained t-value can be lower or higher than the statistical analysis threshold (t-value). If the t-value is lower than the critical value, there is no statistically significant difference between samples, and if it is higher than the critical value there is a statistically significant difference between them. The t-value depends on the means and variabilities of the two datasets (i.e., means, variances, and the number of samples in different scenarios). In this investigation, the t-test is performed for 20 years (60 months for seasonal and 240 months for annual difference). The t-test formula is given in equation 2, where $\overline{X_1}$ and $\overline{X_2}$ are the means, $S_1^2$ and $S_2^2$ are the variances and $n_1$ and $n_2$ are the number of samples,

$$t - value = \frac{|\overline{X_1} - \overline{X_2}|}{\sqrt{\frac{S_1^2}{n_1} + \frac{S_2^2}{n_2}}}$$

For more detail about statistical analysis reader, the readers are encouraged to see (Miller. J. N. and Miller. J. C., 1998).

2) I am also in doubt about the interpretation of the correlations in Fig. 9 and Table 3. If the time-series are not detrended you will mix the signals from trends and from faster variability. The correct way would be to make correlations from detrended series and see if these correlations 'predicts' the size and sign of the trends. I am also in doubt of what and how you conclude about the drivers of the larger reduction in dust in the SAI compared to RCP8.5.

Reply: As requested, detrended annual mean values have been used to calculate the correlation coefficient in Table 3 and Fig. 9. and also, all related sentences are rewritten in the new version of the manuscript.

Minor points:

l20: while for -> except for.

Reply: is implemented.

l150: Is the meaning of the last part of this sentence that you also focus on dust hotspots. I think the sentence is unclear.

Reply: in the new version of the manuscript, these sentences are rewritten.

Fig. 3: In the title to panel f it says 'Seasonal trend'. I think it should be 'Seasonal time-series'. It is not the trend that is shown.

Reply: is implemented.

Fig. 4: In the caption it writes 'd) The annual mean of the dust concentration ..' I think you mean the surface temperature.

Reply: is implemented.

Fig. 5: There are no numbers on the colorbar.

Reply: is implemented.

L209: I think there should be a full stop before 'seasonal cycle plots ..

Reply: is implemented.

I am unsure if the hatching in Fig. 9 shows the significance of the correlations or the regions with large dust concentrations as it says in the caption. The authors should definitely show where the correlations are statistically significant.

Reply: As requested, the statistical analysis test has been performed over the time series, and the significance of the correlation is shown on contour plots using the regions without hatch lines with a 99.5% confidence level. The dashed line contours show dust hotspots (R1 to R5) regions

**Response to the Anonymous Referee #2:**

We thank the reviewers for the comments and suggestions. We think that by implementing the reviewers' comments and suggestions, the revised version has significantly improved.

**Major comments:**

The most surprising result of this work is how little difference there is between the two scenarios RCP8.5 and SAI for most of the parameters considered, Figs. 2, 6, 7, 10, 11; except temperature and TLAI, and temperature is a given considering the added stratospheric aerosol. This is an important point and should be emphasized more.

Reply: Due to the vastness and climatic diversity of the MENA region, it is hard to accurately talk about the annual and monthly time series which averaged across the whole MENA region. However, the third column of Figures 3–8 (i.e., SAI-RCP8.5), show the differences between these two scenarios over the dust hotspots region with more detail. These figures show a 15-50% change for different parameters in the different areas of the MENA region (i.e., up to 15% for dust concentration, up to 30% for surface temperature, up to 35% for leaf area index, up to 20% for wind speed, up to 25% for precipitation, and up to 50% for soil water).

There is ambiguity with the use of the word significant. In statistics it has a very specific meaning, which is used in this paper some of the time. When describing differences it may be used, e.g. a significant difference, to imply a large difference. This is confusing in many places below and only very late in this review did I realize that the authors may be using the first definition, while I was assuming the second.

This leads to the complication of describing differences amongst the scenarios for the different parameters. Words like significant, considerable, large, small, … are subjective and readers and authors may differ on their meaning. In contrast words like larger, smaller, higher, lower, or more preferably percent differences, are objective and not as prone to misinterpretation. That is part of the reason, to suggest in my first review, that the authors present their results in terms of percent difference/change. They did this for some variables, but not all. The whole text should be gone through to make the comparison sentences more objective. Many of the comments below are related to confusion about these words, and disagreements with the authors interpretation/description of the figures.

Reply: As requested, new contour plots are depicted based on the percent of changes, all related sentences are rewritten based on the new figures, and the referee's suggestions are included in them. The percentage of changes has been used to depict the difference between the scenarios, and also all sentences have been rewritten as more objective comparison sentences.

Contour plots for changes in temperature, surface wind speed, and soil water are depicted with the percentage of change related to the current climate. In some areas of the MENA region, precipitation and TLAI parameters have zero or near zero values, and any changes in the future climate will eventuate an infinite positive or negative value for the percentage of changes compared to the current climate. To overcome this problem and for showing the difference based on the percentage of changes, we calculated the percentage of changes relative to the maximum value in the current climate in the MENA region. Also, to investigate dust concentration changes, and to better show the intensity and importance of changes in dust hotspots, the percentage of changes has been calculated compared to the maximum value of dust in the current climate in the MENA region (the maximum value for the mentioned parameters is written in the manuscript to give a sense about the quantity and intensity of change by the reader).

18-21 "This reduction in dust over the MENA region is stronger under the SAI scenario, while for the dry season (e.g., summer with the strongest dust events), more reduction has been projected for the global warming scenario." What is this statement based on, which figures? If anything Fig. 3e, d, g, h) might just suggest the opposite.

Reply: in these sentences, we talked about the higher reduction of dust over most of the dust hotspots, during the summer time. The sentences have been rewritten as below!

Lines 19 - 22: This reduction in dust over the whole MENA region is stronger under the SAI scenario, except over dust hotspots and for the dry season (e.g., summer with the strongest dust events), which more reduction has been projected for the global warming scenario compared with the SAI.

110-115 The amount of annual sulfur injections under the SAI scenario are not specified. Are they known? Text should be added how the SAI scenario is used to maintain 2020 temperature conditions. Is sulfur dioxide injected into the model? Or is the model just artificially nudged to keep the temperature at 2020 levels? If that is the case then aren't the SAI scenarios and the control close to the same, or how do they differ? The authors should explain more fully how the SAI scenario is created.

Reply: The amount of annual sulfur injection and related references are added to the manuscript. Also, the model is not artificially nudged. It is a fully interactive ESM, please see Tilmes et al, 2018 for more detail. Please see more detail about the SAI scenario (e.g., the location of injections, objects of the SAI simulations, the model components, and related references) in the lines 101-122.

180-181. Incomplete sentence. Maybe delete "that".
Reply: is implemented.

183 Northeastern
Reply: is implemented.

Figures 3, 5, ... When gray and white are included in the color scheme, and then stippled over in gray, it is difficult/impossible to separate the color contours from the stippling. For example Fig. 5n), is most of the region the central color, white, or is there some gray factored in? Also on Fig. 5, TLAI, there is no quantitative information on the color bar. What for example indicates no change?
Reply: As mentioned before, the changes are drawn based on the percentage of changes, and these issues have been modified in the figures.

207 "The TLAI under the RCP8.5 scenario shows some significant reduction compared to the CTL across the whole region,..." Here is a case in point from the comment above. What constitutes significant? The gray shades are difficult to separate from the stippling, compare Figs 5d) and e). Without numbers on the color bar how is the reader to know that gray is significant? Why doesn't Fig. 5p) support the above statement. This panel shows that RCP8.5 and CTL are the same except for Jan – May which seems inconsistent with Figs 5g) j), unless the white is not apparent in these figures.
Reply: As requested, all related sentences are rewritten based on the new figures, and the referee's suggestions are included in them. In the new version of the manuscript, the color bar has been modified. Furthermore, as mentioned before, the vastness and variety of the MENA's climate, could be a reason for some inconsistency of the annual and monthly trends with seasonal contour plots. The annual and monthly trends (Fig.10, 11 and S1) over dust hotspots decrease this uncertainty and would give more accurate result for annual and monthly time series (compared with time series for the whole MENA region). The sentences are written as a below.
Lines 224-226: The TLAI under the RCP8.5 scenario shows 5-30 % reduction compared to the CTL across the different area of MENA region, except the region between the Mediterranean and Caspian Seas (Fig. 5a, d, g and j).

214-215 "On the contrary, under the SAI scenario compared to the CTL, the TLAI shows a significant increase both spatially and temporally (Fig. 5b, e, h, k, p and q)." Same problem as above. The color under the stippling (indicating significance) seems to be white, right in the middle of the color bar, which should mean 0/no change, but without a quantitative color bar it

is unknown. The only place with positive colors are Spain and Eastern Europe. Changes shown as percent may help this presentation.

Reply: In the new version of the manuscript, this figure has been replaced with a new one, the color bar has been modified and related sentences have been rewritten according to the new figure presented as a percentage of change.

200-220 TLAI is difficult for any but an expert to understand. Perhaps a few words of explanation would help the reader understand what it means physically. Is it the fraction of a unit area covered by leaves, or??

Reply: the following sentences are included in the manuscript.
Lines: 339- 341: The Leaf area index (LAI) is a quantity to characterize the plant canopies (e.g., the aboveground portion of trees, crops, etc.). LAI is a dimensionless quantity and is defined as $LAI = (one - sided\ leaf\ area\ (m^2)\ /\ ground\ area(m^2))$.

226 "and in the Middle East with two major dust hotspots, the reduction is even stronger" This seems to be splitting hairs for the contour plots. This reader does not see a stronger reduction in the Middle East comparing the left and center set of panels. Perhaps the authors are emphasizing Figs 6p) and q), but calling these differences strong is a stretch. Plotting the differences as a percent would show a small percent difference, barely exceeding the uncertainty bounds.

Reply: As requested, new contour plots are depicted based on the percent of changes, and all related sentences are rewritten based on the new figures, what we meant by saying this sentence is that there is a greater decrease in dust hotspots (shown with black contours) than in its neighboring areas. The sentences rewritten as a below:
Lines 243-245: SAI also shows reductions in the wind speed compared to the CTL during all the seasons, notably in the Middle East with two major dust hotspots, it shows a 5 to 20 % reduction (Fig. 6b, e, h, k, n, p and q).

222-232 A 0.2 m/s wind change is 5% for a 4 m/s wind. Seems pretty close to the uncertainty limit. Still not sure why percent isn't used throughout as suggested in the first review.
Reply: In the new version of the manuscript, this figure has been replaced with a new one (using percent of change), and related sentences have been rewritten according to the new figure please see lines 240-251.

238-241 "The box plot and monthly mean values of the precipitation from different scenarios (Fig. 7p) show that under the RCP8.5 scenario (compared to the CTL), precipitation is projected to significantly increase during the summer season, and under the SAI scenario (compared to the

CTL), this region would experience more precipitation during the spring and summer." It is difficult to see that this statement is supported by Fig. 7p) where the three means of the scenarios are barely outside the 25th/75th quartiles. Perhaps the percent differences are on the order of 10% in some months. Thus why the claim of significant? What do the authors consider is meant by significant, just that statistically the difference of the means is significant, or that physically the difference is significant? The different intent of this word needs to be clarified by the authors.

Reply: In the new version of the manuscript, this figure has been replaced with a new one, and related sentences have been rewritten according to the new figure. The sentences rewritten as a below:
Lines 259-262: The box plot and monthly mean values of the precipitation from different scenarios (Fig. 7p) show that under the RCP8.5 scenario (compared to the CTL), precipitation is projected to almost a 20% increase during the summer season, and under the SAI scenario (compared to the CTL), this region would experience 5 to 25% more precipitation during the spring and summer.

250-254. This language is much improved by discussing the differences as "higher/lower, moderate positive trend", rather than putting a value on the difference, e.g. significant.
Reply: Considering referee suggestion, all sentences have been rewritten as objective comparison sentences.

258 "over the MEAN region (Fig. 9h) and caption Fig. 9", and text following. Where is the MEAN region and why is Figure 9h singled out to illustrate it? Fig. 9 shows the same MENA region shown on Figures 3-8.
Reply: This study was done for MENA region and it was written here as a reminder. This sentence has been corrected in the new version.

260 "columnar dust concentration lower than 35 (µg/m3) are depicted with hatch-line in the Fig 9." Once again the hatching obscures any of the contours except the brightest ones at the ends of the color bars, and in many panels the dust hotspots cannot be identified. Thus the discussion about these hotspots cannot be followed. Plus almost the entirety of each panel is hatched. It would make more sense to hatch the regions where dust concentrations are > 35 ug/m3. Also why is this number chosen? What is its significance?
Reply: This figure has been modified according to the referee's suggestion, and dust hotspot regions are shown using black dashed contour lines in Fig. 9. Also, the statistically significant correlations are shown using statistical analysis with a 99.5% confidence level.

Fig. 9 and Table 3. If the authors wish to compare these two they should be organized in the same way. Table 3 starts with precipitation and ends with wind speed. Fig. 9 is organized in reverse order to Table 3. Plus the dust hotspots are either not included on Fig. 9 or included in such a way that they cannot be seen. This makes it impossible to confirm, or follow the discussion from lines 260-272.

Reply: The order of the variables was modified based on the referee's suggestion.

Fig. 10 The rows of this figure should be organized so that they follow the same pattern as Fig. 9 and Table 3, once they are synced. The order of the parameters at present is somewhat random. Here temperature is at the bottom. In Fig. 9 it is at the top. This just makes more work for the reader to understand the paper.

Reply: The order of the variables was modified based on the referee's suggestion.

274-275. There is almost no change in dust in Fig. 10a3). So how do the authors claim just a lesser than strong reduction here?

Reply: The slight change (approximately 10%) is seen for the SAI scenario and there is no change for RCP8.5. The sentences were rewritten as a below.

Lines 297-300: Although, the dust concentration over the R2 and R3 hotspots has no considerable change by the end of the century for RCP8.5, an approximately 20% and 10% reduction is projected for the SAI scenario over the R2 and R3 hotspots respectively (Fig. 10a2 and 10a3).

283 "considerable reduction of dust concentration between the control and the two future scenarios for both regions" This is a reasonable statement for R5, but not R4. Again stating the differences as percent change is a quantitative statement which is objective. Words like considerable and significant are subjective and can be interpreted many ways.

Reply: in the new version of the manuscript, these sentences are rewritten as a referee's suggestion. please see below lines.

Lines 303 – 305: Figure 11 included error bars for monthly mean values of all considered parameters for R4 and R5 regions, and shows a reduction of dust concentration between the control and the two future scenarios (up to 25%) for R4 and (up to 35%) for R5 regions in spring to fall (Fig. 11a and b).

283-290 The authors frame the discussion as cause and effect. The reduction in wind speed is "controlled by the lower wind speed and higher leaf area index." But that's more than the model can determine. All that can really be said is that they are correlated.

Reply: in the new version of the manuscript, these sentences are rewritten as a below.

Please see lines 303-311: Figure 11 included error bars for monthly mean values of all considered parameters for R4 and R5 regions, and shows a reduction of dust concentration between the control and the two future scenarios (up to 25%) for R4 and (up to 35%) for R5 regions in spring to fall (Fig. 11a and b). Differences between RCP.85 and SAI are however not statistically significant. The monthly mean values with error bars of all considered parameters for R1, R2 and R3 regions are also shown in Fig S1. It seems that the reduction of dust concentration over the R4 region (Fig. 11a) has affected by the lower wind speed (Fig. 11c), and higher precipitation (Fig. 11d) and leaf area index (Fig. 11j) under both RCP8.5 and SAI scenarios. Furthermore, the reduction of the monthly mean value of dust concentration over the R5 region (Fig. 11b) could be a result of the decrease in wind speed (Fig. 11d) and increase in leaf area index (Fig 11j). The results of Fig. 10 and Fig. 11, are in good agreement with the results and correlation coefficients in Table 3.

307-308 "reduction rate of the dust concentration is about 5-40% for the RCP8.5 scenario (compared to CTL), where it is stronger from March to September" Where does 40% come from. The maximum difference in Fig. 2a) is about 4 ug/m3 which is about 16% of the CTL at maximum.

Reply: The stated values are related to the difference between the two scenarios in spring, summer, and autumn for the R4 and R5 dust hotspots (Fig. 3d, g, and j). in the new version of the manuscript, these sentences are rewritten. please see lines 330-334:

The reduction rate of the dust concentration is about 5-35% for the RCP8.5 scenario (compared to CTL), where it is stronger from March to September, especially for the dust hotspots in the Middle-East region (Fig. 3d, g, and j). Similarly, the dust concentration is also found to decline 5-30% under the SAI scenario compared to CTL over the dust hotspots in the MENA region (Fig. 3b, e, h, and k).

314-315 Here the authors suggest the increasing temperature is the reason for decreasing dust, but again all that can really be said is that they are correlated. The cause and effect is difficult to ascertain.

Reply: In the new version of the manuscript, these sentences are removed.

330 "While this increase in the TLAI is found to be small over the dust hotspots" Why is this considered small? TLAI increases in R4/5 by more than 100% and in R2 by this much for SAI.

Reply: it is small compared to the threshold of the TLAI for dust generation. The following sentences are written in the manuscript.

Lines 351-355: Although, more than 100% increase of TLAI is projected in R4 and R5 region for both RCP8.5 and SAI scenarios (Fig. 10e4, e5, Fig. 11i and j), the TLAI mean values over R4 and R5 are still lower than the threshold of the dust emission in the dust generation model (i.e., $TLAI_{mean} < 0.3$). However, this small increase may help to decrease the dust concentrations, since based on correlation coefficients in Fig. 9, and Table.3, there is a negative correlation between the dust and TLAI in these regions.

370-372 "We further conclude that, over the coming 80 years, the dust mass concentration generally decreases with an increase in the precipitation, soil water, and leaf area index, and a decrease in temperature and 10m wind speed over the MENA region" Is this the case for both RCP8.5 and SAI. If so this should be stated.

Reply: In the new version of the manuscript, these sentences are rewritten as a below.

Lines 389-393: We further conclude that, over the coming 80 years, the dust mass concentration generally decreases under the both RCP8.5 and SAI scenarios with an increase in the precipitation, soil water, and leaf area index, and a decrease in 10m wind speed over the MENA region, particularly across the Middle East, and also over the dust hotspots, the near surface wind speed and precipitation have the most impact on this reduction.

---

## Author Response (AR3)

**To**: Editor, ACP

**Subject**: Author Comments of Manuscript, acp-2022-370

**Dear Dr. Jianping Huang,**

Upon the recommendation, we have carefully revised the manuscript entitled "*Future dust concentration over the Middle East and North Africa region under global warming and stratospheric aerosol intervention scenarios*" after considering all the comments and suggestions made by the Referees; all the changes made in the revised manuscript are highlighted in yellow. The following is the point-to-point response to all the comments (the reviewers' comments are rewritten in black color and the replies in blue). We appreciate the opportunity to revise our paper. We believe that the manuscript is much improved after positively addressing all the requested revisions. In the following we provide answer of Anonymous Referee #2.

**Notice:** The line and page numbers refer to the pdf file of "Revised Manuscript".

**Referee #2**

This manuscript can go forward to publication, although there are still problems scattered throughout which should be corrected. The comments below note the ones I found. There may be others. My reading was somewhat quick, but I would not have expected even this much for a third revision. The ellipses below set off quotes from the manuscript.

We thank the reviewers for the comments and suggestions. We think that by implementing the reviewers' comments and suggestions, the revised version has significantly improved.

19 …, except over dust hotspots and for the dry season (e.g., summer with the 20 strongest dust events), which more reduction has been projected for the global warming scenario … This phrase doesn't make sense from "which". End this sentence at … events). Then start a new sentence with whatever the which … is to imply. I can't offer a suggestion as I don't understand it.

Reply: considering the referee's suggestion, the sentences are rewritten as following. Please see lines 19-22: This reduction in dust over the whole MENA region is stronger under the SAI scenario, except over dust hotspots and for the dry season. In other words, in the summer with the strongest dust events, more reduction has been projected for the global warming scenario compared with the SAI scenario.

42 … from remote (what) …?

Reply: this sentence is rewritten.

53 … regional projections over West Africa projected … Maybe try, regional predictions …

Reply: Implemented

62 … models which participated …

Reply: Implemented

198 … Overall, in Fig. 2e, the highest dust concentrations (up to 37 μg/m3) are found … What is the basis for this statement. The scale on Fig. 2e only goes to 30, and none of the lines even get close to this maximum. Oops I see the problem. The labels on the figures are in error. The bottom plot is 2f, not 2e as it is labelled now and led to the comment above.

Reply: Implemented

222-223 Doesn't this suggest that SAI isn't doing anything if there is no change with respect to the control? This should at least be acknowledged if not commented upon.

Reply: as mentioned in the manuscript, geoengineering is considered as the third pillar of climate change policy (alongside mitigation and adaptation efforts) to compensate for anthropogenic warming, and Stratospheric Aerosol Injection (SAI) geoengineering is one of the most discussed strategies. The amount of injection annually adjusts using a feedback-control algorithm to keep a) the global surface temperature, b) interhemispheric and c) equator-to-pole temperature gradients close to the year 2020 conditions (i.e., CTL scenario).

Fig. 5 figure caption … total leaf are …??? Change is misspelled in the figure label for the color bar.

Reply: Implemented

Figs. 5q and r. The authors make no comment on the fact that for TLAI RCP8.5 and CTL are at their maximum now and don't change in the future. While the big difference is that TLAI increases under SAI. This is difficult to understand. Isn't the CTL holding the climate state as it is in 2010 or so? Why isn't this reflected in a difference with RCP8.5, Fig. 5p.

Reply: as stated in the manuscript, the monthly and annual means (Fig. 5p and q) show the trends of mean values over the whole MENA region, for detailed analysis, it is better to see the

spatiotemporal anomalies of TLAI for the different scenarios. Please see lines 224-226: The TLAI under the RCP8.5 scenario shows 5-30 % reduction compared to the CTL across the different area of MENA region, except the region between the Mediterranean and Caspian Seas (Fig. 5a, d, g and j).

260 …The box plot and monthly mean values of the precipitation from different scenarios (Fig. 7p) show that under the RCP8.5 scenario (compared to the CTL), precipitation is projected to almost a 20% increase during the summer season, and under the SAI scenario (compared to the CTL)… These differences are hardly significant. The three scenarios are almost all within their data ranges. Also this may be only true because the precipitation is so low to begin with, making this comment even less significant. The real take home is that for most of the year and throughout the century there is very little difference in the scenarios compared to the CTL.

Reply: although the three scenarios are almost within their data ranges for most of the months, the differences between the SAI and CTL scenarios are obvious for the spring season (i.e., April, May, and June). Also, the differences between the RCP and CTL scenarios are seen in August and September.

264 … MENA region will increase by 5% under the SAI and RCP8.5 scenarios… Is this number even measurable considering the natural variation of precipitation? Again there seems little significant differences between the scenarios. That is the important thing, not small, immeasurable differences.

Reply: the sentences are rewritten based on the referee's suggestion as below, please see lines 263-264: It suggests that the mean annual precipitation across the whole MENA region has little significant differences under the SAI and RCP8.5 scenarios by the end of this century.

266-275 Same comment as above. The differences pointed out are hardly measurable, using Fig. 8q to discuss trends is a stretch, and the surprising thing is how little differences appear between the trends.

Reply: although, the difference in soil water between scenarios is little, it is measurable considering the data distribution and error bar in Fig. 8p and q. Furthermore, the spatiotemporal analysis, reveals the differences between the scenarios obviously.

291- … the wind speed is the main parameter that affects dust concentration change … This seems so obvious that it hardly needs to be verified with a model. Likewise precipitation.

Reply: as you mentioned correctly that is so obvious, nonetheless we want to show the most important factor for dust concentration changes between considered parameters.

Fig. 9 What is the difference between the two columns? One might assume that one is RC8.5 and one SAI but neither the figure caption nor the plots indicate what they are.

Reply: caption and title of the figure is corrected.

Fig. 10 Why isn't the CTL added to these plots so the reader could compare the impact of the two scenarios, not compared to each other, but compared with the current climate. The main take away from Fig. 10 is that aside from temperature and maybe TLAI the two scenarios hardly differ in their impact on all the other parameters. The CTL appears in Fig. 11 so it is available.

Reply: as described in the data and method section and presented in Table 2, CTL stands for the first 20 years (i.e., 2010-2029) of the RCP8.5 scenario. Figure 10 shows the annual mean values of the considered parameters for RCP8.5 (2010-2097) and SAI (2020-2100) scenarios.

299 … approximately 20% and 10% reduction is projected for the SAI scenario over the R2 and R3 hotspots respectively (Fig. 10a2 and 10a3)… Perhaps this is a valid statement for 10 a2, but not for 10 a3 where differences between the scenarios are imperceptible.

Reply: the sentences are rewritten, and the referee's suggestion implemented.

330-335 Or the authors could say that the dust concentrations declined 5-30% under either of the scenarios tested, saving a sentence and drawing attention to the similarities of the two scenarios. But why is this so? According to Fig. 9 dust is correlated with temperature. So why isn't the dust less under SAI since there is a clear temperature difference between these two scenarios.

Reply: as written in lines 333-335, dust concentrations in the summer of the R3, R4, and R5 hotspot regions under the SAI scenario are approximately 5-15% higher than in the RCP8.5 scenario (Fig. 3i). So, there is a weak similarity between two scenarios. Although Fig. 9 shows some correlation between dust and surface temperature, over dust hotspots regions, the correlation coefficient values are small for both scenarios.

337-339 This is the explanation for the previous question that is probably correct and quite interesting. This point should be made early on in the manuscript as it provides a basis for better understanding of the results. And may also contribute significantly to the precipitation changes.

Reply: considering that the mentioned statement is not part of our results, we had to bring this sentence into the discussion section.

341 Shouldn't LAI have already been defined by now. It has been discussed since Fig. 5.

Reply: as the referee suggested, the explanation of LAI moved to the result section, before discussing it in Fig. 5. Please see lines 222-224.

363 … R4 dust hotspot will encounter an enhancement in the annual precipitation (i.e., about 100% and 65% under both RCP8.5 and SAI simulations respectively), consistent with an increase TLAI index (i.e., more than 100% under both scenarios)… Why aren't these differences reflected in Figs. 5 and 7. According to Fig. 7 neither scenario shows more than a few percent difference in precipitation for R4. There is a precipitation increase of 10-20% just north of R4 for some months for SAI, but not RCP8.5. The maximum for the percent difference scale is 25 (30)% for precipitation (TLAI).

Reply: the R4 dust hotspot is located in a semiarid region with lower vegetation cover and precipitation rate compared with the northern part of the MENA region. So, the increases for R4 are not obvious in comparison with the vegetation and precipitation of the southern Europe in Fig. 5 and Fig. 7. However, the annual mean values over the dust hotspots (Fig. 10) reveals the changes more clearly.

372-374. This conclusion seems so obvious that one shouldn't need a model to make that statement.

Reply: as you mentioned correctly that is so obvious, nonetheless we want to show the most important and correlated factors for dust concentration changes between considered parameters.